# A multiplex blood-based assay targeting DNA methylation in PBMCs enables early detection of breast cancer

Tiantian Wang [1,12], Peilong Li [1,12], Qiuchen Qi[1,12], Shujun Zhang[1], Yan Xie[1], Jing Wang[1], Shibiao Liu[1], Suhong Ma[1], Shijun Li[2], Tingting Gong[3], Huiting Xu[4], Mengqiu Xiong[5], Guanghua Li[6], Chongge You[7], Zhaofan Luo[8], Juan Li [1] ✉, Lutao Du [1,9] ✉ & Chuanxin Wang [1,10,11] ✉

The immune system can monitor tumor development, and DNA methylation is involved in the body's immune response to tumors. In this work, we investigate whether DNA methylation alterations in peripheral blood mononuclear cells (PBMCs) could be used as markers for early detection of breast cancer (BC) from the perspective of tumor immune alterations. We identify four BC-specific methylation markers by combining Infinium 850 K BeadChips, pyrosequencing and targeted bisulfite sequencing. Based on the four methylation markers in PBMCs of BC, we develop an efficient and convenient multiplex methylation-specific quantitative PCR assay for the detection of BC and validate its diagnostic performance in a multicenter cohort. This assay was able to distinguish early-stage BC patients from normal controls, with an AUC of 0.940, sensitivity of 93.2%, and specificity of 90.4%. More importantly, this assay outperformed existing clinical diagnostic methods, especially in the detection of early-stage and minimal tumors.

Breast cancer (BC) has surpassed lung cancer to become the most common malignancy in the world, with an estimated 2.3 million new cases and approximately 685 thousand deaths each year, posing a serious threat to human health[1]. Early diagnosis is the key to prolonging survival time and reducing cancer mortality, with the relative reduction degree ranging from 15 to 25% in randomized trials[2,3]. Mammography and ultrasound are the main methods of early screening for BC in clinical practice, but both have limitations[4-6]: the sensitivity of mammography in the diagnosis of dense BC is as low as 30%, and the diagnostic accuracy of ultrasonography depends on the examination equipment and physician's experience; thereby, the false positive rate of imaging examination is high, which can easily lead to over-diagnosis and over-treatment[7,8]. Therefore, to improve the prognosis and outcome of the disease, it is urgent to develop a

[1]Department of Clinical Laboratory, The Second Hospital of Shandong University, 247 Beiyuan Street, Jinan 250033 Shandong, China. [2]Clinical Laboratory, The First Hospital of Dalian Medical University, Dalian 116011, P. R. China. [3]Clinical Laboratory, The First Affiliated Hospital of Anhui Medical University, Hefei 230022, P. R. China. [4]Departmemt of Clinical Laboratory Medicine, Affiliated Tumor Hospital of Nantong University, 226361, Jiangsu, China; Medical School of Nantong University, Nantong 226001, P. R. China. [5]Clinical Laboratory, Nanjing First Hospital, Nanjing Medical University, Nanjing 210006, P. R. China. [6]Department of clinical laboratory, Guangdong Provincial People's Hospital/Guangdong Academy of Medical Sciences, Guangzhou 510000, P. R. China. [7]Laboratory Medicine Center, Lanzhou University Second Hospital, the Second Clinical Medical College of Lanzhou University, Lanzhou 730000, P. R. China. [8]Department of Clinical Laboratory, The Seventh Affiliated Hospital of Sun Yat-Sen University, Shenzhen 518107, P. R. China. [9]Department of Clinical Laboratory, Qilu Hospital of Shandong University, Shandong Provincial Key Laboratory of Innovation Technology in Laboratory Medicine, Jinan 250012, P. R. China. [10]Shandong Engineering & Technology Research Center for Tumor Marker Detection, Jinan 250033, China. [11]Shandong Provincial Clinical Medicine Research Center for Clinical Laboratory, Jinan 250033, China. [12]These authors contributed equally: Tiantian Wang, Peilong Li, Qiuchen Qi. ✉e-mail: niderouke@163.com; lutaodu@sdu.edu.cn; cxwang@sdu.edu.cn

highly sensitive and specific detection method for the early diagnosis of BC.

DNA methylation is an important epigenetic modification, which can regulate gene expression without altering DNA sequence. It plays an important role in many biological processes, including the occurrence and development of cancer[9,10]. In particular, hypermethylation of the CpG island promoter may lead to transcriptional silencing of tumor suppressor genes, which significantly affects the process of tumorigenesis. DNA methylation changes exist in almost all cancers and occur in precancerous or early cancer stage[11–14]. Thus, it is considered to be an ideal marker for early diagnosis of cancer. Previous studies that explored the potential of DNA methylation as a cancer biomarker have focused on tumor tissue and tumor-derived materials including circulating tumor cells[15], circulating tumor DNA[16], cell-free DNA[17,18], and tumor-host microenvironment[19]. However, tumor tissues are relatively difficult to obtain and have strong invasiveness; circulating tumor cells can be easily mixed with normal cells, which causes low specificity; circulating tumor DNA in serum or plasma has a low abundance and high possibility of fragmentation. These defects substantially limit their clinical applications. Recent studies have demonstrated that DNA methylation in peripheral blood cells has the potential to serve as complementary cancer biomarker[20,21]. Cancer-specific alterations in DNA methylation have been identified in peripheral immune cells from various cancer types, including hepatocellular carcinoma (HCC)[22], prostate cancer[23], colorectal cancer[24], and head and neck squamous cell carcinoma[25]. There is a significant difference in the DNA methylation profile of peripheral blood mononuclear cells (PBMCs) between chronic hepatitis and HCC[22], suggesting the epigenetic reprogramming of the host immune system during the development of malignant tumors. However, whether DNA methylation status in PBMCs could be used to detect BC progression is still unclear.

In this study, we performed a genome-wide DNA methylation profiling for BC and normal controls using 850k BeadChips to identify the BC-specific methylation landscape in PBMCs. After validating the identified methylation markers via pyrosequencing and targeted bisulfite sequencing (TBS), we developed a multiplex quantitative methylation-specific PCR assay based on selected methylation markers for the early detection of BC (BC-mqmsPCR). Finally, we systematically evaluated the performance of BC-mqmsPCR in the diagnoses of early-stage BC and minimal BC tumors in a multicenter cohort, and compared it with traditional tumor markers CA153, CA125, and CEA.

## Results

### Study design and participants

A total of 820 patients were enrolled in this study from ten hospitals in six provinces of China between May 2020 and July 2022. The cases were selected by being (i) Pathological diagnosis of BC (without any treatment such as surgery and radiotherapy and chemotherapy); (ii) No other cancers were present; (iii) No other known inflammatory diseases (bacterial or viral infections, asthma, autoimmune diseases, active thyroid disease) that may alter PBMCs characteristics; (iv) Subjects can understand and sign written informed consent to participate in the study. Controls were free of malignant diseases and were frequently matched to cases of age and race. All methylation tests were conducted in PBMCs samples. Because blood samples were collected prior to pathological diagnosis and clinical treatments, 39 samples were excluded from analysis due to: (1) the lack of pathological data; (2) pathologically confirmed benign lesions; (3) insufficient amount of DNA extracted from PBMCs; (4) low-quality tests. The remaining 781 PBMCs samples (366 BC, 290 normal controls, and 125 other tumors) were used for DNA methylation profiling, methylation marker screening, and the development of early BC diagnostic methods. An overview of the study design is shown in Fig. 1. The clinical characteristics and demographics of the 781 patients and the training and validation cohort patients are summarized in Supplementary Data 1 and 2, respectively.

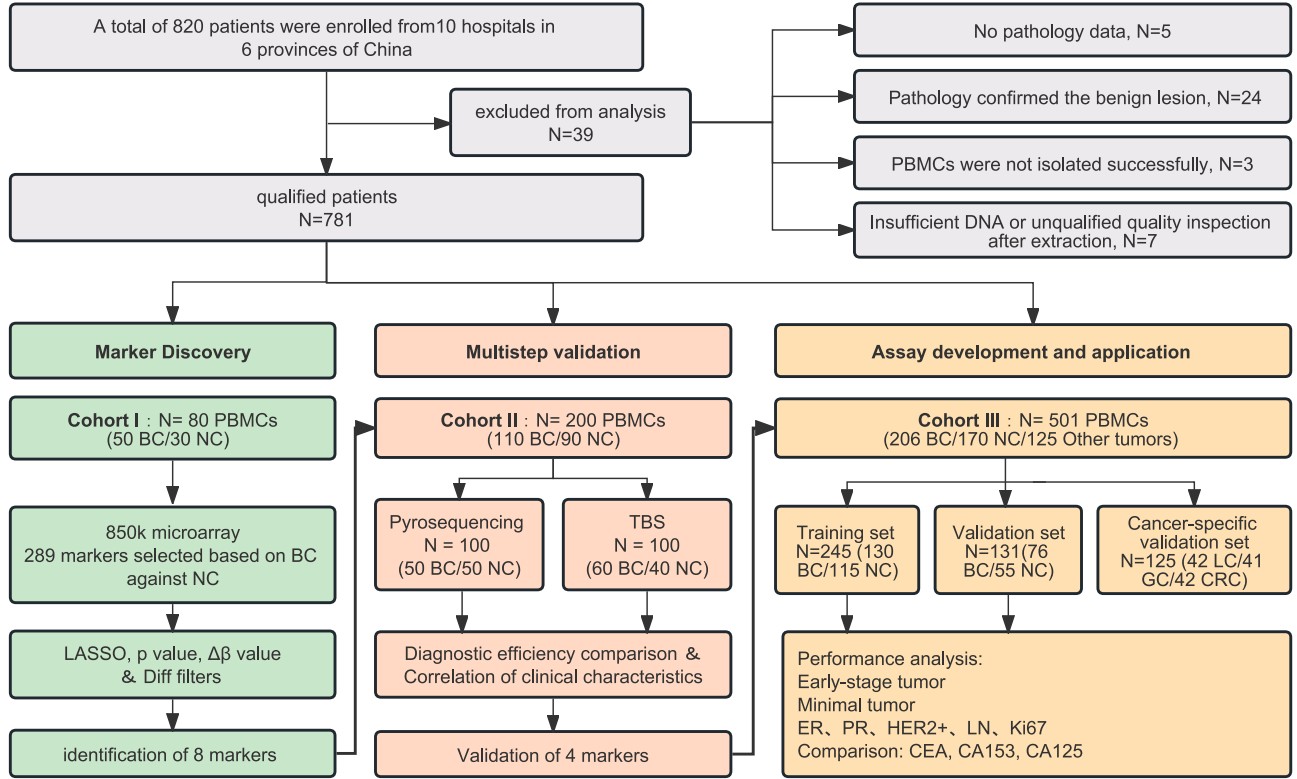

**Fig. 1 | Workflow chart of the study design.** BC breast cancer, NC normal controls, PBMCs peripheral blood mononuclear cell, LASSO the least absolute shrinkage and selection operator, ER estrogen receptor, PR progesterone receptor, LN Lymph node metastasis, LC lung cancer, GC gastric cancer, CRC colorectal cancer.

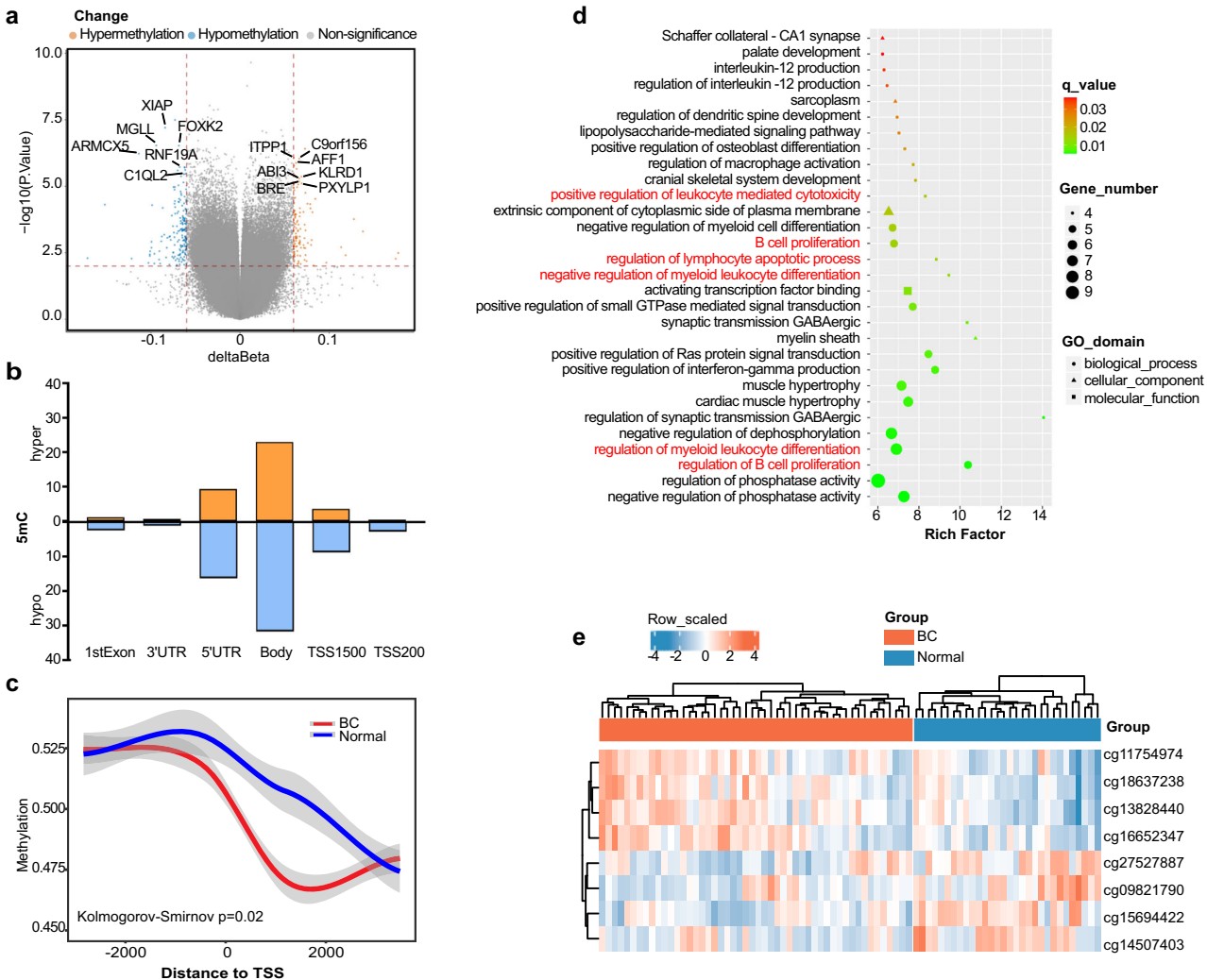

**Fig. 2 | The landscapes of BC-associated DNA methylation signatures in PBMCs by Infinium 850 K microarray. a** The volcano plot for differentially methylated CpG positions (DMPs) between BC patients and normal controls. The x-axis is the mean DNA-methylation difference (delta Beta), and the y-axis is the log10 of the $P$ value. Hyper-methylated CGs are shown in orange, hypomethylated CGs are shown in blue, and non-significant methylated CGs are shown in gray. **b** Bar charts representing the relative distribution of DMPs. Orange and blue bars represent hypermethylation and hypomethylation events, respectively. **c** Distribution of upstream and downstream methylation of TSS in PBMCs of BC patients and normal controls. $P$ values were determined by two-sided loess regression. TSS, transcription start site. **d** The GO enrichment results of differentially methylated genes showed that multiple items were involved in the immune system and immune surveillance system. The size of the dot represents the number of differential genes, the color of the dot represents whether the q value is significant, and the trait of the dot represents different GO classification. **e** Unsupervised hierarchical heatmap clustering of eight candidate methylation markers that are differentially methylated between BC samples ($n = 50$) and normal controls ($n = 30$). Source data are provided as a Source Data file.

In the Discovery phase, we performed a genome-wide DNA methylation profiling assay in an independent cohort including 50 BC patients and 30 normal controls (Cohort I, $n = 80$), aiming to establish a DNA methylation landscape and identify BC-specific methylation markers in PBMCs. Next, the identified methylation markers that can be used for BC diagnosis were validated in cohort II ($n = 200$, 110 BC and 90 normal controls), including the pyrosequencing set ($n = 100$, 50 BC and 50 normal controls) and TBS set ($n = 100$, 60 BC and 40 normal controls). Finally, a multicenter cohort III ($n = 501$, 206 BC, 170 normal controls and 125 other tumors), which was divided into training and validation sets was used for the development and evaluation of the BC-mqmsPCR BC diagnostic assay.

### Genome-wide DNA methylation analysis reveals the PBMCs DNA methylation landscape for BC

To explore BC-specific DNA methylation markers in PBMCs, we used Infinium Human Methylation 850 K BeadChip to examine the genome-wide DNA methylation status in PBMCs from 50 BC patients and 30 normal controls. After normalization and batch calibration, a total of 820,000 markers were identified for subsequent analysis. PCA plot shows little difference between BC and normal control (Supplementary Fig. 1a, b). The CpG positions with $|\Delta\beta| \geq 0.06$ and $p$ value $\leq 0.01$ were defined as the differentially methylated CpG positions (DMPs)[26–28]. There were 289 DMPs (194 genes) between BC and normal controls, among which 112 DMPs (38.8%) were significantly hyper-methylated, and 177 DMPs (61.2%) were significantly hypomethylated in tumors (Fig. 2a). This result is consistent with a previous study on overall 5mC changes in cancer, which identified a low number of hypermethylation events among a large number of hypomethylated changes[29]. The genomic distribution of DMPs showed that hypo-DMPs and hype-DMPs accounted for 31.6 and 23.0% of the gene body region, respectively, and hypomethylation events were more frequent, which is consistent with gene bodies being more heavily methylated in normal cells (Fig. 2b). The distribution of DMPs varied among

chromosomes (Supplementary Fig. 1c). Furthermore, the transcription start sites (TSS) of BC patients were hypomethylated compared with normal controls (Fig. 2c), which may be related to the immune activation in BC patients. This finding is consistent with the results from ref. 30. The differentially methylated genes were then subjected to pathway analysis. Notably, genes associated with DMPs were enriched in the pathways related to immune surveillance and immuno-editing systems (Fig. 2d and Supplementary Fig. 1d). These results support the hypothesis that DNA methylation changes in PBMCs of BC patients are associated with host immune responses.

### Identification of DNA methylation markers in PBMCs to distinguish BC patients from normal controls

Next, we applied a series of screening principles to identify the most important and specific DNA methylation markers of BC. By using LASSO analysis, 33 methylation markers were obtained (Supplementary Table 1), and using |Δβ| ≥0.08, p value ≤0.0001 screening principle (based on the Champ DMP function), eight methylation markers were obtained (Supplementary Table 2), with six methylation markers overlapped in the two screening criteria. After removing two methylation markers (cg21723696 and cg14928964) that were difficult to design with primers, we ultimately selected four overlapping methylation markers (cg14507403, cg09821790, cg15694422, and cg27527887) and two methylation markers (cg11754974 and cg16652347) that only met |Δβ| ≥0.08, p value ≤0.0001 screening principle for further validation. Given the notion that methylation in the TSS region has been shown to be more important than other CpG sites in regulating gene expression[31], we included two more methylation sites (cg13828440/KLRD1 and cg18637238/KLRK1) in the TSS region that might be associated with the development of BC. Varker et al. showed that cancer-related stress in patients with invasive BC is related to immune impairment. In patients exhibiting high levels of cancer-related psychological stress, NK cytotoxicity is reduced and CD94 (KLRD1) levels are reduced[32]. Starčević et al. confirmed that the regulation of NK cell activation in tumor invasion may be involved in the pathogenesis of BC. The expression of peripheral NK cell-activated receptors (CD94/NKG2C, NKG2D, and CD16) and inhibitory receptors (NKG2A) from BC patients was reduced, highlighting the importance of NK cells as suitable targets for effective antitumor responses in the BC immunosuppressive tumor microenvironment[33]. Kruijf et al. confirmed that NKG2D ligand (NKG2DL) is often highly expressed in BC, and NKG2DL can bind to the activating NKG2D (KLRK1) receptor present on NK cells and subsets of T cells, thus initiating the immune response and playing a crucial role in tumor immune editing in BC[34]. Other studies have linked NKG2D to the development of BC[35–37]. Therefore, a total of eight candidate markers were selected for the final validation, including four hyper-methylated markers and four hypomethylated markers (Table 1 and Figs. 2e, 3a).

To rule out the influence of cell mixture distribution in PBMCs on DNA methylation, we calculated the cell proportion according to EpiDISH method[38], and found significantly higher monocytes proportion in BC patients than in controls, and NK cells proportion in BC were significantly lower than those in normal controls, but not in CD8+ T cells, CD4+ T cells, B cells and granulocytes (Supplementary Table 3). Nevertheless, adjusted for the cell proportions, the eight markers still showed significant difference between BC and normal controls (Supplementary Table 4).

The reproducibility of these 8 markers was evaluated using pyrosequencing and TBS in a cohort of 200 patients. The results showed that there was no significant difference in 4 hypomethylated markers (cg14507403, cg09821790, cg15694422, and cg27527887) between BC and normal controls (Supplementary Fig. 2a); while the other 4 hypermethylated markers exhibited stable higher methylation level in BC patients compared to normal controls in both pyrosequencing and TBS (Fig. 3b). Notably, the trend of methylation difference at site cg16652347 and cg13828440 in TCGA is diametrically opposite to that in PBMCs, which proves that the methylation difference we detected comes from PBMCs, not BC tissue (Supplementary Fig. 2b). Since all the data in TCGA were sequenced by 450 K chip, excluding the two sites (cg11754974 and cg18637238) that we screened in 850 K chip, it is impossible to evaluate the methylation differences between these two sites in BC and adjacent normal tissues.

Interestingly, we found that in the pyrosequencing set, the methylation level of cg18637238 was negatively correlated with tumor size, with significantly higher methylation in small tumor (≤1.5 cm). There were no significant differences in age, lymph node metastasis status, ER status, PR status, HER2 status, or Ki67 levels (Supplementary Fig. 2c). In the TBS set, the methylation levels of cg11754974, cg13828440, and cg18637238 were higher in BC with stage 0/I or II than those in stage III/IV, and there was no obvious difference regarding tumor size, age, lymph node metastasis status, ER status, PR status, HER2 status, or Ki67 levels (Fig. 3c). Next, to characterize the potential of these methylation markers in detecting early BC and small-size BC tumors, we established the receiver operating characteristic (ROC) curves. As expected, each methylation marker could distinguish early-stage BC and small tumors (≤1.5 cm) from normal controls in both sets (Fig. 3d and Supplementary Fig. 2d). The unsupervised hierarchical clustering of these 4 markers was able to distinguish BC from normal controls with moderate specificity and sensitivity (Fig. 3e, Supplementary Fig. 2e, and Supplementary Table 5).

### Development and application of BC-mqmsPCR Assay

To detect multiple markers in a rapid, cost-effective, and more convenient way in clinics, we developed an efficient method to diagnose BC, BC-mqmsPCR, which allows simultaneous multiplex quantification of four methylation markers in a single reaction[39]. Two different

**Table 1 | Features of eight differentially methylated CpG positions among breast cancer and normal controls group**

| Probe ID | Δβ value (BC-NC) * | P Value | Adjust P Value | CHR | UCSC_RefGene | UCSC_RefGene_Group |
|----------|---------------------|---------|----------------|-----|--------------|---------------------|
| cg14507403 | −0.151771106 | 4.92E-05 | 0.047018976 | 7 | LMTK2 | Body |
| cg09821790 | −0.09869485 | 5.12E-05 | 0.047837259 | 16 | SLC7A6 | Body |
| cg15694422 | −0.094113112 | 2.92E-07 | 0.01187436 | 3 | MGLL | Body |
| cg27527887 | −0.086760967 | 6.52E-05 | 0.05261645 | 10 | / | / |
| cg11754974 | 0.081256084 | 9.46E-05 | 0.057515784 | 14 | TRDJ3 | TSS1500 |
| cg16652347 | 0.084377724 | 2.97E-05 | 0.039173184 | 7 | PLXNA4 | Body |
| cg13828440 | 0.068921275 | 4.34E-06 | 0.021774546 | 12 | KLRD1 | TSS1500 |
| cg18637238 | 0.061870615 | 2.68E-05 | 0.037570473 | 12 | KLRK1 | TSS1500 |

BC breast cancer, NC normal controls, CHR chromosome, TSS1500 1500 bp of the start site of transcription.

*The Δβ value (BC-NC) is the average β values of the BC group minus the average β values of the normal controls group. The differential methylated CpGs position were calculated by champ.DMP function. The adjusted p value were computed using the Benjamini–Hochberg method. Probe ID: identification of the probe (from 850 K BeadChip Array).

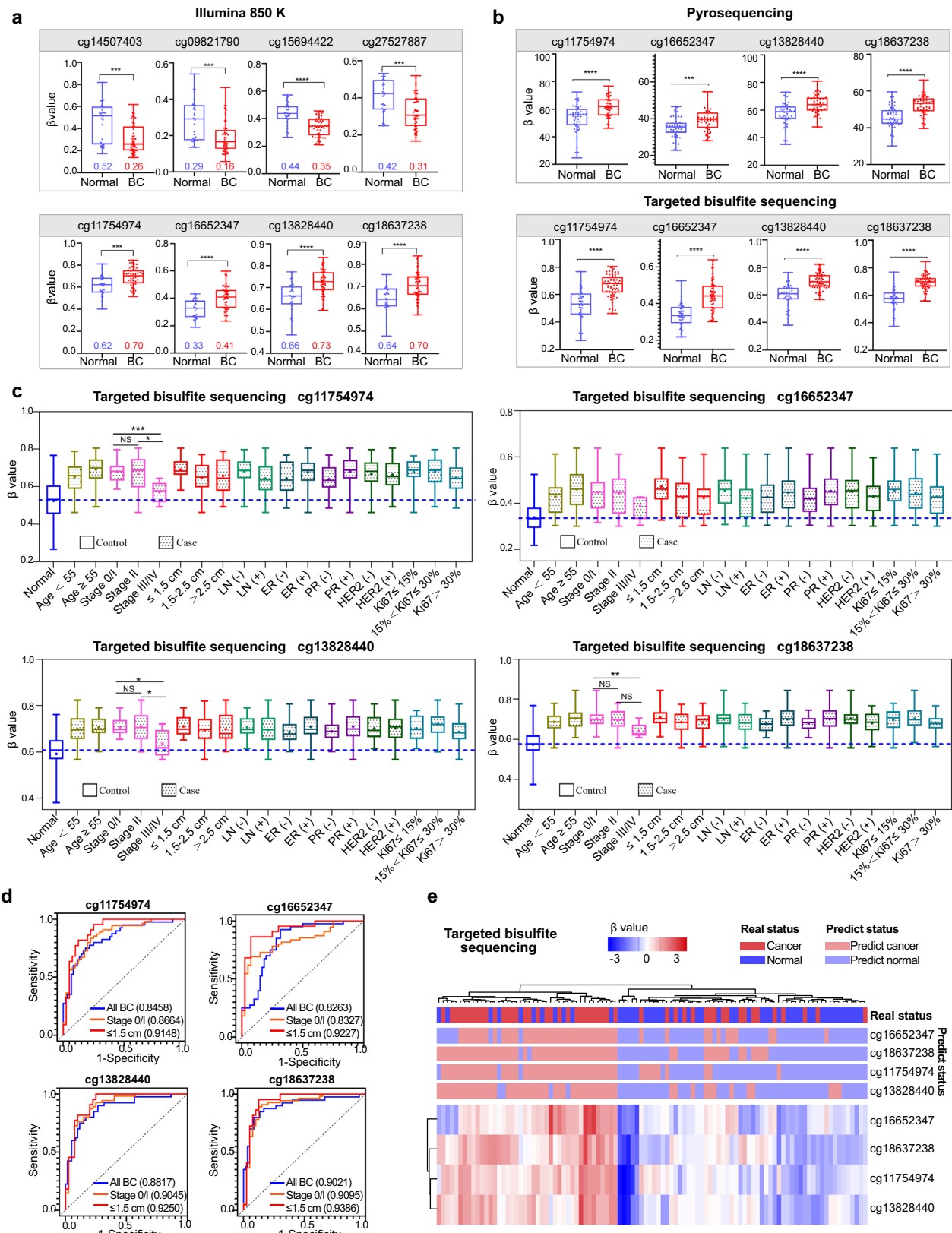

fluorophore probes were used for BC-mqmsPCR detection. FAM fluorophore was selected to label 4 methylation markers, and VIC fluorophore was used for ACTB reference control. The ΔCT value represented the methylation level (ΔCT = CT_reference − CT_biomarker)[40]. To validate the feasibility of BC-mqmsPCR assay, we compared the 4-marker BC-mqmsPCR assay with individual marker qMSP assay using bisulfite-transformed PBMCs DNA from the same case. The BC-

mqmsPCR assay produced a ΔCT value which was 2.22, 5.95, 0.71, and 1.76 higher than the individual cg11754974, cg16652347, cg13828440, and cg18637238 assays, respectively, confirming that multiplex assay could achieve fluorescence signal accumulation and is analytically more sensitive than the individual assays (Fig. 4a). To validate whether BC-mqmsPCR assay could be used to distinguish BC patients from normal controls, we performed 4-marker BC-mqmsPCR

**Fig. 3 | The identification and validation of BC-associated DNA methylation markers. a** The box plot shows the β-value distribution of eight candidate methylation markers in PBMCs from BC patients ($n$ = 50) and normal controls ($n$ = 30). A β-value of zero represents no methylation, and 1 represents full methylation. $P$ values were determined by two-sided Mann–Whitney $U$-test. ***$p$ ≤ 0.001; ****$p$ ≤ 0.0001. **b** The box plots present the β-value distribution of four methylation markers validated by pyrosequencing (top, 50 BC and 50 normal controls) and targeted bisulfite sequencing (bottom, 60 BC and 40 normal controls) in PBMCs samples. $P$ values were determined by a two-sided Mann–Whitney $U$-test. ***$p$ ≤ 0.001; ****$p$ ≤ 0.0001. **c** The methylation levels of four methylation markers in 40 normal controls and 60 BC patients of the targeted bisulfite sequencing set with different age, stage, tumor size, lymph node metastasis status, ER status, PR status, HER2 status, and Ki67 levels (one-Way ANOVA test, two-sided, Dunnett's test for multiple comparisons). *$p$ ≤ 0.05; ***$p$ ≤ 0.001; NS no significance. **d** ROC curves of four methylation markers in all BC, stage 0/I BC and ≤1.5 cm BC of the targeted bisulfite sequencing set. The AUCs for the different categories are shown in the legend. **e** Unsupervised hierarchical clustering of four methylation markers based on differential methylation levels between BC patients and normal controls in the targeted bisulfite sequencing set. The boxes in **a–c** are bounded by the first and third quartile with a horizontal line at the median; minima is the smallest data greater than or equal to the first quartile − 1.5 × interquartile range (IQR); maxima is the largest data point less than or equal quartile + 1.5 × IQR. Source data are provided as a Source Data file.

assay and individual marker qMSP assay to detect the methylation levels of 40 BC patients and 40 normal controls (Derived from the training set in Cohort III). As shown in Supplementary Fig. 3a, b, the methylation level was higher in BC patients than that of normal controls in both BC-mqmsPCR assay and qMSP assay, but the AUC of 4-marker BC-mqmsPCR assay was significantly higher than that of each individual marker qMSP assay (0.925 vs 0.811, 0.785, 0.760, and 0.793), suggesting that the diagnostic performance of qMSP could be improved by combining individual markers. Although using LASSO analysis to weigh individual markers can also improve performance to the level of multiple analyses (AUC = 0.94). However, this study focused on finding a more convenient, fast, and economical detection method. Multiple analysis can not only meet the clinical needs of rapid and convenient clinical detection, but also has high diagnostic efficiency. Then, we further evaluated the analytical sensitivity of the BC-mqmsPCR assay and each individual marker qMSP assay by mixing 50 ng/ul of bisulfite-transformed PBMCs DNA with pure water and diluting it to 50, 10, 1, 0.1, 0.01, and 0.001%. The result showed that, the quantitative detection limit of BC-mqmsPCR assay was 0.01% (Fig. 4b), which was much lower than the 1% for cg11754974 (Supplementary Fig. 3c), and 0.1% for cg16652347, cg13828440, and cg18637238 (Supplementary Fig. 3d–f), suggesting that BC-mqmsPCR was a more sensitive assay.

## BC-mqmsPCR analysis of PBMCs DNA for BC diagnosis

To further assess the performance of BC-mqmsPCR assay for clinical application, we recruited a multicenter cohort of 206 BC patients and 170 normal controls from ten hospitals in China and divided into training and validation sets. Training set (130 BC and 115 normal controls) were recruited from six centers for methodology development and search for the optimal cut-off value, and validation sets (76 BC and 55 normal controls) were recruited from another four centers for external validation. The results showed that the methylation level of the BC group was significantly higher than that of the normal controls group in both training and validation sets (Fig. 4c, d). We also examined the performance of BC-mqmsPCR, which exhibited AUCs (0.925 and 0.918), sensitivity (83.1 and 80.3%), and specificity (90.4 and 89.1%) for training and validation sets (Fig. 4e–h). More importantly, after determining the positive and negative classification of BC-mqmsPCR by the cut-off value of Youden's index (3.223) in the training set, we found that the predicted results of BC-mqmsPCR had a high consistency with the pathological diagnosis results (Fig. 4i, j). In addition, a total of 125 patients with other cancers, including 42 colorectal cancer, 42 lung cancer, and 41 gastric cancers, were enrolled to evaluate the diagnostic specificity of BC-mqmsPCR for BC. The results showed that BC-mqmsPCR could distinguish BC from other non-BC patients with an AUC of 91.3% (Fig. 4k, l).

## The application of BC-mqmsPCR in detecting early-stage BC

Early diagnosis of BC is the key to reducing patient mortality. Given that individual markers showed good diagnostic value in minimal tumor and early-stage BC, we further evaluated the performance of BC-mqmsPCR in early-stage BC. As shown in Fig. 5a, the methylation level was significantly higher in stage 0/I and II BC patients than stage III patients, and BC-mqmsPCR exhibited AUCs (0.940 and 0.927) and sensitivities (93.3 and 85.7%) for the training and validation sets (Fig. 5b, c and Supplementary Fig. 4a). Importantly, the BC-mqmsPCR can diagnose the minimal tumors (≤1.5 cm), with AUCs up to 0.945 and 0.936, and sensitivities up to 93.2 and 90.0% in the training and validation sets, respectively (Fig. 5d, e and Supplementary Fig. 4a). These results suggest that the BC-mqmsPCR assay is a supplementary method for the identification of small tumors at an early stage, which is difficult to identify by ultrasound and mammogram examination.

The potential utility of this approach was highlighted in two cases (from the training set of Cohort III) that were diagnosed by BC-mqmsPCR but missed by ultrasound, mammogram and serum tumor markers CEA, CA153, and CA125 (Fig. 5f). Patient #1 was admitted to the hospital due to the presence of left breast nodules in physical examination for 3 years. Breast ultrasound showed three hypoechoic nodules in the left breast with clear borders and inhomogeneous internal echogenicity, which was consistent with BI-RADS grade 3 (Negative) (Fig. 5g). The mammogram showed that the upper left breast gland was slightly denser than the contralateral one, and a round-like calcified foci were seen in the lower left breast quadrant, which was consistent with BI-RADS grade 2 (Negative) (Fig. 5h). However, BC-mqmsPCR assay diagnosis was BC positive, which was consistent with the postoperative pathology showing low-grade ductal carcinoma in situ with a tumor resection area of 1.5 × 0.6 cm (Fig. 5i). A similar situation was observed in another case, which was diagnosed as BC with BC-mqmsPCR and pathologically confirmed as medium/high-grade ductal carcinoma in situ, but missed by ultrasound and mammogram (Supplementary Fig. 4b–d). These results strongly demonstrate the advantages of BC-mqmsPCR for the detection of minimally sized tumors and early BC.

## BC-mqmsPCR outperformed traditional methods of BC diagnosis

We compared the performance between BC-mqmsPCR and currently used tumor markers CA153, CEA, and CA125 in a cohort of 67 patients with early-stage BC (stage 0–1 BC patients with complete CEA, CA153, CA125, and BC-mqmsPCR information in the training set and validation set) and 113 patients with small to medium-sized mass BC (tumor size ≤2.5 cm BC patients with complete CEA, CA153, CA125, and BC-mqmsPCR information in the training set and validation set), respectively. The clinical characteristics and diagnostic performance of the 4 methods are shown in Fig. 5j. Surprisingly, BC-mqmsPCR detected 15 out of 17 stage 0 BC cases (88.2%) that were mainly characterized by carcinoma in situ, while only one case was identified by CEA (1/17, 5.9%), and none were detected by CA153 (0%) or CA125 (0%). Because the traditional tumor markers CEA, CA153, and CA125 may only be used in patients with advanced BC, they are not helpful for screening or diagnosis of early BC[41,42]. For example, CA153 concentrations are increased in 10% of patients with stage I disease, 20% with stage II disease, 40% with stage III disease, and 75% with stage IV disease[43,44], so

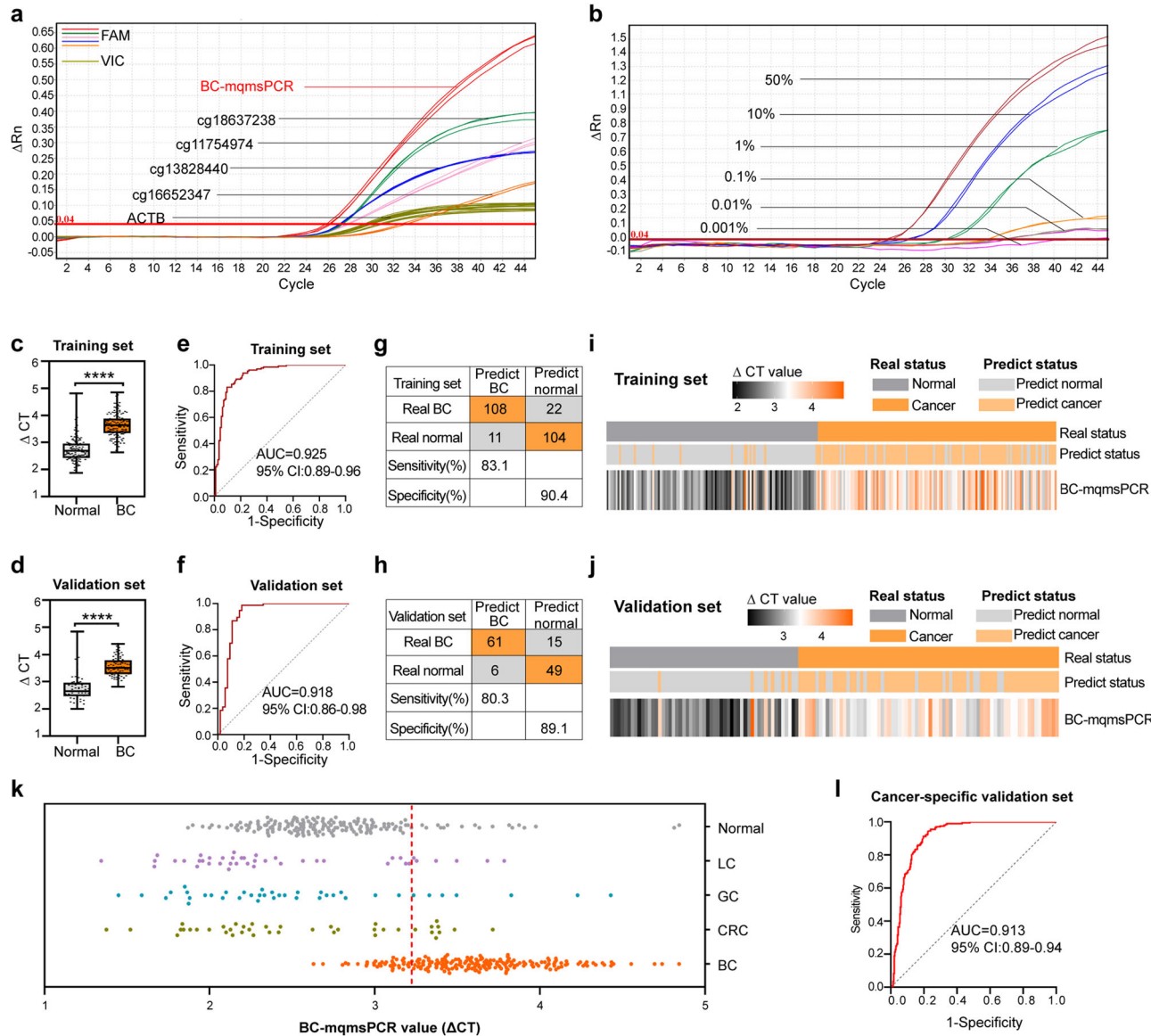

**Fig. 4 | Development and application of BC-mqmsPCR for BC diagnosis.**
**a** Comparison between the BC-mqmsPCR and uniplex qMSP assays. The BC-mqmsPCR assay produced a ΔCT values that were higher than cg11754974, cg16652347, cg13828440, and cg18637238 by 2.22, 5.95, 0.71, and 1.76, respectively. FAM represents the DNA methylation signal, and VIC represents the internal reference control signal. The value of ΔRn represents the amount of degradation of the probe during the PCR process, which is the amount of PCR product. ΔRn=R + n - R-n, where R + n represents the fluorescence intensity measured at each point, and R-n represents the fluorescence baseline intensity. **b** Assessment of analytical sensitivity of BC-mqmsPCR assay. The BC-mqmsPCR assay detected PBMCs DNA signals with as little as 0.01% of 50 ng total PBMCs DNA diluted with pure water. **c**, **d** Methylation levels of PBMCs DNA as quantified by BC-mqmsPCR assay in the training set (130 BC patients and 115 normal controls) and validation set (76 BC patients and 55 normal controls). The y-axis represents methylation levels ($\Delta CT = CT_{reference} - CT_{biomarker}$), in which a higher value represents a higher methylation level. $P$ values were determined by a two-sided Mann–Whitney $U$-test. ****$p \le 0.0001$. **e**, **f** The AUC of BC-mqmsPCR was 0.925 (0.89–0.96) for the training set (**e**) and 0.918 (0.86–0.98) for the validation set (**f**). **g**, **h** Confusion tables of the binary results of BC-mqmsPCR in the training set (**g**) and validation set (**h**). **i**, **j** Supervised hierarchical clustering of the differentially methylated BC-mqmsPCR signature between BC patients and normal controls in the training set (**i**, $n = 245$) and validation set (**j**, $n = 131$). **k** Distribution of BC-mqmsPCR value in BC ($n = 206$), LC ($n = 42$), GC ($n = 41$), CRC ($n = 42$) and normal controls ($n = 170$). The red dashed line represents the cut-off value (3.223). LC lung cancer, GC gastric cancer, CRC colorectal cancer. **l** The AUC for distinguishing BC from non-BC was 0.913. The boxes in **c**, **d** are bounded by the first and third quartile with a horizontal line at the median; minima is the smallest data greater than or equal to the first quartile – 1.5 × interquartile range (IQR); maxima is the largest data point less than or equal quartile + 1.5 × IQR. Source data are provided as a Source Data file.

the detection rate of traditional tumor markers in early BC is very low. In addition, BC-mqmsPCR showed higher sensitivity (91.7%) for detecting BC patients with small nodules (≤1.5 cm) than CA153 (0%), CA125 (2.1%), and CEA (0%). The overall detection rate of BC-mqmsPCR was 82.0%, which was significantly higher than CA153 (5.3%), CEA (6.0%), and CA125 (3.5%) (Supplementary Fig. 4e and Supplementary Table 6). Importantly, the diagnostic sensitivity of BC-mqmsPCR in CA125, CEA and CA153 negative BC patients was 82.42, 81.41, and

81.99%, respectively (Supplementary Fig. 4f). Collectively, our results showed that the established BC-mqmsPCR assay had the potential to serve as a clinical diagnostic approach, especially for early-stage BC patients.

DNA methylation alterations occur at the beginning of tumor growth. A previous longitudinal study on 200,000 people showed that a DNA methylation-based blood test can detect five types of cancer at four years before conventional diagnosis[45]. In addition, a study led by

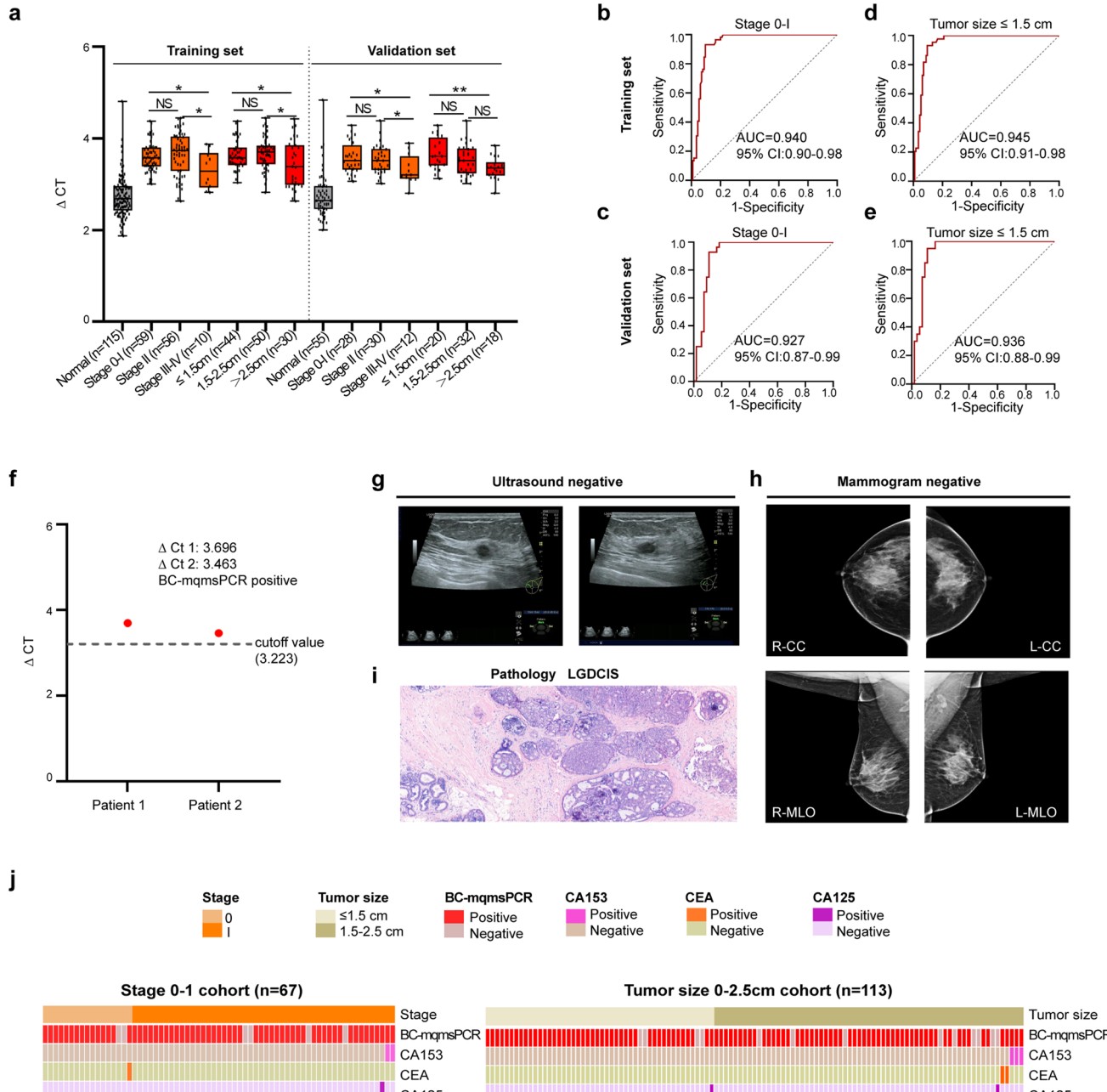

**Fig. 5 | The application of BC-mqmsPCR in the detection of minimal tumor and early-stage BC. a** The methylation levels of PBMCs DNA as quantified by BC-mqmsPCR assay in breast cancer samples of different stages and tumor sizes in training and validation sets (one-Way ANOVA test, two-sided, Dunnett's test for multiple comparisons). *$p \leq 0.05$; **$p \leq 0.01$; NS no significance. **b**, **c** The AUC of BC-mqmsPCR for detecting stage 0-I BC was 0.940 (95%CI:0.90–0.98; n = 175) in the training set and 0.927 (95%CI:0.87–0.99; n = 83) in the validation set. **d**, **e** The AUC of BC-mqmsPCR for detecting minimal BC (≤1.5 cm) was 0.945 (95%CI:0.91–0.98; n = 159) in the training set and 0.936 (95%CI:0.88–0.99; n = 75) in the validation set. **f** Examples of two patients, who were later diagnosed by pathological biopsy, were detected with minimal tumors by BC-mqmsPCR, but missed by ultrasound, mammogram, and serum tumor markers CEA, CA153, and CA125. **g**–**i** Ultrasound,

mammogram, and pathology results of patient 1. The pathology of the tumor was low-grade ductal carcinoma in situ (LGDCIS) with a tumor resection area of 1.5 × 0.6 cm. The magnifications of the Hematoxylin and eosin (H&E) staining images in pathology were ×200. **j** The distribution of predicted diagnostic status using BC-mqmsPCR in patients with early-stage BC (stage 0–1 cohort, n = 67) and minimal BC (tumor size ≤2.5 cm, n = 113), along with the information of tumor stage, size, CA153, CEA, and CA125 results. The boxes in a are bounded by the first and third quartile with a horizontal line at the median; minima is the smallest data greater than or equal to the first quartile − 1.5 × interquartile range (IQR); maxima is the largest data point less than or equal quartile + 1.5 × IQR. Source data are provided as a Source Data file.

University College London found that the *EFC#93* DNA region of BC samples showed abnormal DNA methylation patterns that could be used to diagnose tumors up to one year earlier than existing screening methods[46]. Therefore, we conducted a follow-up study on 170 normal control patients in the training (115 normal controls) and validation sets (55 normal controls), and found that among the 17 patients who

were predicted to be positive (2 cases were lost to follow-up), two were diagnosed with BC within 1 year of follow-up; moreover, among the 153 negative patients (22 cases were lost to follow-up), no BC was confirmed within 1 year of follow-up (Supplementary Fig. 5a, b and Supplementary Data 3). We speculate that the above results may be caused by the following reasons. First, some patients with positive BC-

mqmsPCR test may have cancerous changes in their body, but the current clinical detection methods can't detect it, until the tumor develops to a certain size, it will be confirmed by clinical conventional detection methods; Secondly, the current clinical diagnosis of BC mainly relies on imaging examinations, while the existing imaging examinations have their limitations (high false negative in patients with dense breast, depending on the level of clinicians, etc.), so patients wrongly grouped due to false negative imaging examination cannot be excluded. Thirdly, the detection method of BC-mqmsPCR also has certain false positives. Cross-contamination of target sequences or amplification products is an important factor leading to false positives, among which aerosol contamination is the most common. Cross-contamination of PCR reagents and samples can also lead to false positives. In this regard, we should standardize the laboratory design and complete different experimental operations in continuous independent space. Regular ventilation and disinfection, and the use of aerosol remover. Use disposable utensils or autoclaved consumables to avoid cross-contamination. With the extension of follow-up time, these patients will not develop BC.

Therefore, we recommend that patients with no risk factors for BC who test positive for BC-mqmsPCR should be reviewed with breast ultrasound or mammography every 6 months, and then 12 months and 24 months later. If the lesion remains stable, it can be rechecked every 1–2 years. Patients with high-risk factors for BC who test positive for BC-mqmsPCR should be reviewed every 3 months. They were then reexamined at 6, 12, and 24 months. If the lesion remains stable, it can be rechecked every 1 year thereafter.

## Discussion

BC remains to be the leading cause of cancer-related death for women worldwide, and an important clinical challenge of treating BC is the lack of noninvasive hematologic biomarkers to detect early-stage tumors. Although the screening methods based on mammography can help reduce the mortality rate of BC by 28–45%, its application in young patients is limited by its high radiation, poor discrimination for dense breast lesions, and the possibility of false negative diagnosis in patients with small breasts[47,48]. In this study, we analyzed PBMCs DNA methylation profiles in BC patients and developed an efficient method, named BC-mqmsPCR, based on four selected PBMCs DNA methylation markers. The BC-mqmsPCR method showed high sensitivity and specificity for the diagnosis of early and minimal BC.

In recent years, DNA methylation has become the basis of many biomarkers for cancer diagnosis and prognosis because they are more stable than other candidate biomarkers, such as RNA or protein-based markers. Although DNA methylation is usually tissue and cell type specific, recent studies have shown that the epigenetic changes in DNA of peripheral blood immune cells can also serve as potential biomarkers for solid tumors[20,49,50]. In 1909, immunologist Paul Ehrlich[51] hypothesized that the host immune system could recognize and destroy early tumors, and thereby the abnormal immune function could be one of the basic causes of tumorigenesis. The subsequent immune surveillance theory proposed by Burnet and Thomas[52], as well as the tumor immune editing theory proposed by Gavin P Dunn and Robert D Schreiber[53] further confirmed the relationship between immunity and tumor, which were also experimentally confirmed[54]. PBMCs contain multiple members of the host immune surveillance system that control tumorigenesis, so it can be used as a source of cancer biomarkers.

In this study, we analyzed the DNA methylation changes in PBMCs samples from BC patients and normal controls by 850k microarray. A total of 289 BC-associated DMPs were identified, most of which were hypomethylated DMPs (61.2%); Differences in DNA methylation profiles might be influenced by the proportions of the PBMCs composition. According to our Illumina 850 K data, the proportion of Monocyte in BC patients was higher than that in normal controls, and the proportion of NK cells was lower than that in normal controls, while the proportion of CD8+ T cells, CD4+ T cells, B cells, and granulocytes was similar between BC patients and normal controls. After adjusting for cell proportion, the eight markers we screened still showed significant differences between BC and normal controls, suggesting that changes in the proportion of PBMCs subsets were not the reason for the observed difference in methylation in PBMCs, and that changes in methylation profile may be the activation of the host immune responses during BC progression. The differentially methylated genes were enriched in immune functions[55–57], such as regulation of B cell proliferation, regulation of myeloid leukocyte differentiation, and regulation of lymphocyte apoptotic process, supporting the hypothesis that DNA methylation changes in PBMCs of BC patients are associated with host immune responses. Eight methylation markers were selected for multi-phase validation, and cg18637238, cg16652347, cg13828440, and cg11754974 passed the validation. The CpG sites cg18637238, cg13828440, and cg11754974 are located in the TSS1500 regions of KLRK1, KLRD1, and TRDJ3 genes, respectively. These genes are related to immune receptors[58]. Among them, KLRK1 mediates the antitumor functions of NK cells, as well as conventional and unconventional T cells[59], and the presence of KLRD1/NKG2A on human tumor-specific T cells impairs IL2 receptor-dependent proliferation[60,61]. The CpG site cg16652347, located in the gene body of PLXNA4, is involved in the production of neutralizing cytokines from immune cells and is induced upon T-cell activation; thus, it can be used to monitor the patient's response to immune checkpoint blockade[62,63]. These results support the hypothesis that changes in DNA methylation in PBMCs are associated with the immune system of the host organism and reflect the epigenetic reprogramming process of the immune system in BC. Notably, unlike cfDNA and ctDNA methylation studies, which usually have low diagnostic sensitivity for early-stage cancers[64,65], our study found that the diagnostic efficacy of DNA methylation markers in PBMCs was higher in a cohort of early-stage and minimal BC than in a BC cohort of all stages and sizes. We speculate that the reason may be that the immune system is activated in the early stage of the tumor and suppressed in the late stage[66].

Pyrosequencing and TBS have the advantages of high accuracy, real-time quantification, and automatic detection for methylation. However, these methods also have high cost, low detection throughput, and inability to detect multiple genetic markers, which limit their clinical application. To overcome these limitations, we developed the BC-mqmsPCR method to simultaneously detect the above four methylation markers in a single tube. The BC-mqmsPCR assay is more sensitive than the single-marker qMSP assay and has better diagnostic efficacy for BC. In a multicenter cohort containing 206 BC patients, the AUC of BC-mqmsPCR was as high as 0.921, and it showed poor diagnostic efficiency for lung cancer, colorectal cancer and gastric cancer, suggesting that it is a BC-specific diagnostic method.

At present, several blood-based DNA methylation biomarkers have been proposed for early detection of BC, such as RPTOR, MGRN1, and RAPSN, with a combined AUC of 0.79 in validation cohort I, and 0.60 and 0.62 in validation cohorts II and III, respectively[27]. By performing bisulfite sequencing PCR, Shan et al. showed that RASSF1a, P16, and PCDHGB7 could be used as diagnostic biomarkers for BC, and the combination of these three genes showed an AUC of 0.781[67]. The methylation signature identified in our study had an AUC of 0.925, with high sensitivity of 83.1% and high specificity of 90.4%. It is worth noting that BC-mqmsPCR has a better diagnostic performance for early-stage BC and minimal tumors, and it can detect 15 out of 17 cases (88.2%) of stage 0 BC. In tumors with diameter ≤1.5 cm, the diagnostic sensitivity of BC-mqmsPCR is higher, which is up to 93.2% in the training set and 90.0% in the validation set. Moreover, BC-mqmsPCR can also identify tumors earlier than existing clinical diagnostic methods.

However, our study still has some limitations. First, subjects in this study excluded inflammatory diseases that might alter PBMCs characteristics (bacterial or viral infections, asthma, autoimmune diseases, active thyroid diseases), so it is not possible to assess whether the diagnostic tests developed in this study are currently available for patients with inflammatory diseases such as auto-immune diseases. Second, this study is a retrospective study, and the results need to be validated in prospective study cohorts. Third, we only followed on healthy cohort for a relatively short period of time, so the effect of longer follow-up on disease outcome could not be determined; moreover, due to the lack of follow-up data on BC patients, we couldn't study the effect of these methylation changes on recurrence or patient survival. Finally, the mechanism of candidate DNA methylation changes is still unclear, and their impact on the occurrence and progression of BC needs to be further explored. In the next study, we will further explore the relevant issues that cannot be analyzed in this study.

In conclusion, we developed a simple and efficient BC-mqmsPCR assay based on four DNA methylation markers in PBMCs for rapid and noninvasive diagnosis of BC. This method has high sensitivity and specificity, especially for the detection of early-stage BC and minimal tumors. In addition, BC-mqmsPCR can also identify tumors earlier than existing clinical procedures, showing the potential to further improve early diagnosis of BC and patient prognosis.

### Ethical statement
The corresponding author, on behalf of all authors, jointly and severally, certifies that their institution has approved the protocol for any investigation involving humans and that all experimentation was conducted in conformity with ethical and humane principles of research.

## Methods
### Patient recruitment and sample collection
We conducted a multicenter retrospective study using the PBMCs samples collected from ten hospitals in six provinces in China. From May 2020 to July 2022, a total of 820 patients were enrolled in the study. The participating hospitals were: the Second Hospital of Shandong University, Qilu Hospital of Shandong University, the 960th Hospital of the Chinese People's Liberation Army, the First Hospital of Dalian Medical University, the First Hospital of Anhui Medical University, the Fifth People's Hospital of Nantong, the Second Affiliated Hospital of Lanzhou University, the Seventh Hospital of Sun Yat-sen University, the First People's Hospital of Nanjing, and the Guangdong Provincial People's Hospital. Sample collection was approved by the ethical committees of each hospital, and all participants signed the written informed consent.

All cases were initially selected based on the following criteria: (1) newly diagnosed BC patients, (2) no other tumors, (3) no auto-immune diseases, acute inflammation, or other pathogenic factors that affect the methylation status of PBMCs. All samples were collected at the time of diagnosis before any treatment, including chemotherapy or radiotherapy. The control group included age-matched females with normal breast ultrasound or mammography examination, and with no obvious abnormalities in blood examination, no other tumors, and the above pathogenic factors that affect the methylation status of PBMCs. We also collected the untreated blood samples from patients with other tumors ($n = 125$, 42 lung cancer, 41 gastric cancer, and 42 colorectal cancer) to verify the specificity of early diagnostic markers for BC. Samples from all cases were processed in the following way: 1–2 ml of peripheral blood was collected in EDTA vacuum tubes, and PBMCs were isolated from blood within 2 h by density gradient centrifugation using Ficoll solution (Sigma-Aldrich, Histopaque-1077) and stored at −80 °C until DNA extraction.

### DNA extraction and bisulfite conversion
PBMCs DNA was extracted with the TIANamp Genomic DNA Kit (Cat# DP304-03), according to the manufacturer's instructions. DNA purity and concentration were estimated using a Quibt3.0 fluorometer (Thermo Fisher Scientific); then, 2 µl of DNA samples were run on a nucleic acid gel for inspection. Low-quality tests (insufficient purity of PBMCs DNA after extraction, such as A260/280 ≤ 1.6 or ≥2.1, protein or RNA contamination, and cannot be accurately quantified) were removed. The extracted DNA was stored at −80 °C until further use. For bisulfite conversion, 1 µg of PBMCs DNA were processed with the Zymo EZ DNA Methylation-Gold Kit according to the manufacturer's instructions. Briefly, 130 µL of Lightning Conversion Reagent was added to 20 µL DNA sample; the samples were then incubated in a thermocycler with the following program: 98 °C for 10 min, 64 °C for 150 min, and 4 °C for ∞. After that, the bisulfite-converted DNA was mixed with M-Binding buffer, run through a Zymo-SpinTM IC Column, desulphonated, washed, and eluted in 20 µL M-Elution buffer.

### Infinium human methylation 850 K BeadChip
The infinium human methylation 850 K BeadChip analysis was performed according to the manufacturer's instructions, and the data were analyzed using the ChAMP 2.18.2 package in R 4.0.0. β value was used to represent DNA methylation level [β = intensity of the methylated allele (M)/(intensity of the unmethylated allele (U) + intensity of the methylated allele (M) + 100)], and it was expressed as a continuous variable that ranged from 0 (no methylation) to 1 (full methylation). First, we filtered out the probes with detection $p$ value > 0.01, probes with <3 beads in at least 10% of samples, non-CpG probes, multi-hit probes, probes located on chromosome Y, and SNP-related probes[68,69]. A total of 820,000 probes were obtained for subsequent analysis. Then β value matrix were normalized using BMIQ for adjusting type I and type II probe bias. Next, we used SVA (singular value decomposition analysis) to analyze the batch effect caused by BeadChip Slide and Array, and then we applied Combat to correct this batch effect. All CpG sites were annotated using EPICanno.ilm10b4.hg19, and the differential methylated CpGs position (DMPs) were calculated by champ.DMP function. The adj.$p$ values were computed using the Benjamini–Hochberg method. CpGs having |Δβ| ≥0.06 and $p$ value ≤0.01 were considered as DMPs ($p$ value has not been corrected by multiple tests).

### Gene ontology (GO) analysis and kyoto encyclopedia of genes and genomes (KEGG) analyses
To clarify the biological functions of the genes and the involved signaling pathways, we annotated each Gene based on the GO and KEGG database. Enrichment calculations were performed using Fisher's exact test. Further, we also need to conduct GO and pathway enrichment analysis of the genes. The specific principle is to carry out annotation mapping of differentially expressed genes in GO and KEGG database entries, calculate the number of the genes in each GO and pathway entry, and then use a hypergeometric test for statistics. Select the GO and KEGG entries that are significantly enriched in the differentially expressed genes. After the calculated $p$ value was corrected by multiple hypothesis tests, the $p$ value 0.05 was taken as the threshold, and the GO and KEGG term meeting this condition was defined as the GO and KEGG term significantly enriched in the target genes, Rich Factor, whose calculation formula is: (diff_gene_in_this_pathway/diff_gene_in_all_pathway) (all_gene_in_this_pathway/all_gene_in_all_pathway). The biological processes of GO and KEGG pathways enrichment analyses were carried out by clusterProfiler package of R 4.0.0. The figure was drawn by ggplot2.

### Selection of candidate methylation markers
Candidate methylation markers were selected according to LASSO, Δβ, $p$ value, and location. Using the LASSO method (10-fold cross-

validation), 33 methylation markers were obtained, and using |Δβ| ≥0.08, *p* value ≤0.0001 screening principle, eight methylation markers were obtained, with six methylation markers overlapped in the two screening criteria. After removing two methylation markers (cg21723696 and cg14928964) that were difficult to design with primers, we ultimately selected four overlapping methylation markers (cg14507403, cg09821790, cg15694422, and cg27527887) and two methylation markers (cg11754974 and cg16652347) that only met |Δβ| ≥0.08, *p* value ≤0.0001 screening principle for further validation. Given the notion that methylation in the TSS region has been shown to be more important than other CpG sites in regulating gene expression, we included two more methylation sites (cg13828440 and cg18637238) in the TSS region that might be associated with the development of BC. Finally, eight candidate markers were selected for further validation.

### Pyrosequencing

Pyrosequencing was performed using a PyroMark Q48 kit (Qiagen) following the manufacturer's protocol. PCR primers and Pyro primers were designed using PyroMark Assay Design Software 2.0 (Supplementary Table 7). The generated pyrograms were checked for sequencing quality in terms of bisulfite conversion efficiency, peak shape, peak height, and the sequence fidelity. The absolute methylation levels of CpG sites were calculated by Pyro Q-CpG Software[70] that calculates the ratio of converted to unconverted cytosines at each CpG. Δβ was used to compare the magnitude of the mean difference between the experimental group and control group, and the two-tailed Mann–Whitney *U*-test was used to compare whether the difference between the two groups was significant. The significance threshold was set at *P* < 0.05.

### Targeted bisulfite sequencing

Targeted bisulfite sequencing (Genesky Biotechnologies Inc, Shanghai, China) is a multiplex-targeted CpG methylation analysis method based on next-generation sequencing technology. Genomic DNA was subjected to sodium bisulfite treatment using the EZ DNA methylation kit according to the manufacturer's protocol. Multiplex PCR was performed using an optimized primer combination (Supplementary Table 8). After PCR amplification and library construction, samples were sequenced on an Illumina MiSeq (Illumina, CA, USA) platform. Sequencing was performed using a 2 × 150 bp terminal pairing pattern on double-ends, and the methylation level of each CpG site was expressed as the percentage of methylated cytosine in total cytosine.

### Expression of candidate methylation markers in BC and adjacent normal tissues in the TCGA database

In order to confirm the expression of four methylation markers screened from PBMCs in BC and normal tissues in TCGA database, we downloaded 450k Illumina Infinium methylation arrays sequencing data of 794 BC tissues and 96 adjacent normal tissues from the TCGA database, and found that only two CpG sites cg16652347 and cg13828440 existed in the 450k methylation arrays sequencing data. Then, we compared the differences of DNA methylation expression in cg16652347 and cg13828440 in BC tissues and adjacent normal tissues by using a rank-sum test.

### qMSP and BC-mqmsPCR assay

All primers and probes were designed using MethPrimer with human curation when necessary (Supplementary Table 9), and then synthesized by GE Biological Technology (Jiangsu, China). The PCR products were TA-cloned into pGEM-T Easy vector and sequenced. For qMSP reactions, cg11754974, cg16652347, cg13828440, or cg18637238 was analyzed together with the β-actin (ACTB) control assay in the same reaction using FAM and VIC-based probes, respectively. The BC-mqmsPCR assay contains 4 qMSP assays targeting cg11754974, cg16652347, cg13828440, and cg18637238 with the FAM-based probes,

along with the same ACTB control assay using VIC-based probe. The KAPA PROBE FAST qPCR Master Mix (2×) Kit was used for qMSP and BC-mqmsPCR assay. The reactions were performed in a 10-μL system with 5 μL of 2× Master Mix, 0.2 μL of ROX Low, 0.25 μM of each target primer, 0.1 μM of each target probe, 0.06 μM of each ACTB primer, 0.05 μM of ACTB probe, and 40 to 50 ng of bisulfite-converted DNA. PCR cycling conditions were as follows: heat activation at 95 °C for 3 min, followed by 45 cycles of 95 °C for 3 s and 58 °C for 30 s. The ΔCT value was used to represent the methylation level, $\Delta CT = CT_{reference} - CT_{biomarker}$.

### Follow up process

We conducted telephone follow-up on 170 normal controls included in the training and validation sets. Firstly, we defined the time point of BC-mqmsPCR detection after collecting samples from normal controls as Test time 0. According to the cut-off value of 3.223, of 170 normal controls, 17 predicted positive and 153 predicted negative. Then we followed up every 6 months for 2 years on average. The last follow-up time minus the collection time was defined as the follow-up time of each sample. During the follow-up, a pathological examination was carried out for those with abnormal imaging examination. According to the pathological results, it was confirmed whether the patient had developed BC. No pathological examination was performed in patients with no abnormal imaging findings during follow-up.

### Reporting summary

Further information on research design is available in the Nature Portfolio Reporting Summary linked to this article.

## Data availability

Genome-wide DNA methylation data from the discovery phase is publicly available in the Gene Expression Omnibus (GEO accession No: GSE237036) The original data presented in graphs generated in this study are provided in the Supplementary Information and Source Data file. Source data are provided with this paper.

## Code availability

The codes used to generate the analysis and figures in this study can be accessed at http://www.bioconductor.org/packages/release/bioc/vignettes/ChAMP/inst/doc/ChAMP.html?from=singlemessage&isappinstalled=0#section-.

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

## Acknowledgements

We are grateful to the patients and healthy volunteers who participated in this study. We also thank Professor Zhang Tao of Shandong University for his statistical support. This work was supported by the Key Research and Development Program of Shandong Province (2021ZLGX02 and 2020CXGC011304 to C.W.), the National Natural Science Foundation of China (82272412 and 82002229 to J.L.), Taishan Scholars Climbing Program of Shandong Province (NO.tspd20210323 to C.W.), Young Taishan Scholars Program of Shandong Province (NO.tsqn201909176 to L.D., NO.tsqn202211322 to J.L., NO.tsqn202211323 to P.L.), Outstanding Young and Middle-aged Scholar of Shandong University, Qilu Young Scholars Program of Shandong University, Shandong University Clinical Research Project (2020SDUCRCA002 to C.W.), and Tumor Biomarker Innovation Team Foundation of Jinan City (2021GXRC020 to L.D.).

## Author contributions

J.L., L.D., and C.W. designed research; T.W., P.L., Y.X., J.W., S.L., S.M., S.L., T.G., H.X., M.X., G.L., C.Y., and Z.L. enrolled patients; T.W., S.Z., Q.Q., and P.L. performed research; J.L., L.D., C.W., T.W., P.L., Y.X., J.W., S.Z., Q.Q., and S.L. analyzed data; T.W. and P.L. wrote the paper.

## Competing interests

The authors declare no competing interests.
