## [Peer Review File · Nature Communications]

REVIEWER COMMENTS

Reviewer #1 (Remarks to the Author): expertise in machine learning methods

Summary

In this work, Wang et al. introduce a blood-based methylation assay to detect breast cancer from circulating PBMCs. The rationale is early detection of malignancies from non-invasive procedures would be highly beneficial to treatment efficacy and patient outcomes, and builds on existing work in the field using methylation profiles from blood for cancer detection. Here the authors create a well-powered cohort including 820 patients from across 10 hospitals, that are split into three cohorts ("marker discovery", "multistep validation", "assay development and application") that use increasingly fine-grained methylation identification (850k microarray, then pyro/targeted bisulphite sequencing, then a new "BC-mqmsPCR" assay). The BC-mqmsPCR assay is created to simultaneously quantify four methylation markers in a single reaction. The paper identifies four markers that are hypermethylated in breast cancer that validate across the cohorts, and show impressive sensitivity / specificity (92.7%, 85.7%) for early detection of breast cancer. They go on to show that this discriminates from other common cancer types, has higher predictive capacity than existing blood based methylation markers of BC, and may help detect BC cases missed on existing imaging diagnostic systems.

Strengths

- Paper tackles important clinical problem, with state-of-the-art results
- The dataset used in the study is impressive, with a large sample size and patients from multiple hospitals, which increases the generalizability of the findings.
- The paper is well written and organized, making it easy to follow the research design and results.

Weaknesses

- The biological interpretation of the methylation markers identified is confusing and hard to contextualize with the state-of-the-art biomarker power
- General lack of detail in the computational analyses used to create the diagnostic test
- The train/validation split of the data is mixed across hospitals, which means that an unbiased estimate of the performance of the diagnostic test in a hospital that did not have training data is missing.

Major

The sample and patient selection in the study raises some questions/requires clarification:

- The study states that "low quality tests were removed" but it is unclear what criteria were used to define a "low quality" test.
- The lack of pathological data for some patients raises concerns about whether there could be a correlation between this missing data and the absence of cancer. Can the authors comment on this?
- The methods section of the paper presents additional information on patient selection criteria (such as no other tumors or autoimmune diseases), but this information should be highlighted earlier in the paper, alongside the sample selection criteria, and discussed as a major limitation of the study. This is because the diagnostic test developed in the study cannot currently be used in patients who have an autoimmune disease or other tumors.

The resulting biomarkers identified raise some questions about what is actually being measured. 3/4 of the markers measured are only really expressed in T/NK cells (human protein atlas). Consequently, it's really hard to understand how this test measures anything other than T/NK cell burden in the blood since KLR genes aren't specific to breast cancer, but NK cells in general. This interpretation is somewhat backed up by the fact that the authors find the opposite pattern of methylation in TCGA (vs normal tissue) and that the discriminatory power of the test decreases as

tumour size increases (you would somewhat expect the opposite as a larger tumour would shed more / illicit more of an immune response). While this doesn't invalidate the use of this probe as a biomarker, it makes understanding what's actually happening really difficult. I would be interested in the other reviewers' thoughts in this matter, and would encourage the authors to look at the predictive power of their biomarkers in an entirely independent cohort (e.g. Tang et al. Oncotarget 2016 as referenced in their paper).

The computational analysis performed requires additional details:

- Normalization strategy - SVA and combat are used for normalization, does this happen across the train / validation cohort simultaneously? If so this would count as train/test leakage. Similarly, what (if any) normalization procedure is used for the BC-mqmsPCR assay?
- GO and Kegg enrichment preformed "using R package". Which R package? What parameters?
- It's written for pyrosequencing "Absolute methylation levels of CpG sites were calculated by software". Which software and how are they calculated?

The initial set of methylation sites were chosen via various thresholds (e.g. $p < 0.0001$), but this only returns 6 so an extra 2 are added "because they were located in the 1500bp of the TSS region and closely related to the BC onset". This requires more details. What is the rationale for ad-hoc adding in extra genes? TSS region for which gene and how were these genes chosen? How was this chosen? What paper shows a link to BC onset? Were these 2 genes part of the final 4 that are hypermethylated and validate, or hypomethylated and don't validate?

It is not sufficiently detailed what training actually happens in the training/validation step for BC-mqms stage:

- Is this to find a threshold that maximizes the Youden index which is then applied to the validation set?
- How are the four ΔCT variables (for each of the four selected CpG sites) combined into a single score?
- Figure 4K presents a "BC-mqmsPCR Probability score". This is not referenced anywhere else (and doesn't range between 0 and 1 like a probability should).

The training and validation cohorts mix across hospitals. In other words, the AUC scores presented are equivalent to the performance expected if there was always training data collected from the hospital in which the test will be deployed. Can the authors perform an analysis where training/validation are stratified across hospitals to quantify the performance expected if training data was not available from a given hospital?

On line 383 authors write that other methylation biomarkers have been proposed for early BC detection (e.g. RPTOR) and state some AUCs on cohorts. Can the authors clarify if the AUCs / cohorts listed are their own or the original papers' results? If original papers, can they quantify the predictive power of these markers in their data?

Minor

- Figure 1 typo "pathology"
- Missing citation(s) for "DNA methylation exists in almost all cancers and occurs in precancerous or early cancer stage"

Reviewer #2 (Remarks to the Author): expertise in DNA methylation in breast cancer

In this study, Wang et al. used Infinium 850K BeadChips to identify changes in DNA methylation in peripheral blood mononuclear cells (PBMCs) between patients with breast cancer (BC) and healthy controls. Using pyrosequencing and targeted bisulphite sequencing, Wang et al. validated four of the BC-specific PBMCs markers and developed, based on the four markers, a multiplex methylation-specific quantitative PCR assay for the detection of BC. Wang et al. validated the performance of this assay in a multicenter cohort and highlighted its value for the diagnosis of

early-stage BC. This study is very interesting and timely. While the validation of the performance of the BC-mqmsPCR assay appears convincing, the identification and validation of the BC-specific PBMCs markers is compromised by several major weaknesses.

Major comments:

1. The description of the cohorts, especially the subsampling of the different cohorts throughout the paper, is confusing and should be clarified. For example:

o Line 236-237: "...qMSP assay to detect the methylation levels of 40 BC patients and 40 normal controls." -> To which cohort belong these samples? Please indicate in the text.

o Line 269-270: "...the methylation level was significantly higher in stage 0/I and II BC patients than stage III patients..." -> Where do these samples come from and how many were there for each stage? Please clarify the cohort as well as the sample number per stage in text and figures (Fig. 1 & Fig. 5A)

o Line 296-297: "...in a cohort of 67 patients with early-stage BC and 113 patients with small to medium-sized mass BC..." -> Where do these samples come from and how many were there for each stage? Please clarify the cohort as well as the sample number per stage in text and figures (Fig. 1 & Fig. 5J)

o Line 316-317: "... follow-up study on 170 healthy control patients in the training and validation sets..." -> To which cohort belong these samples? Please indicate in the text.

o Line 278-280: "...two cases that were diagnosed by BC-mqmsPCR but missed by ultrasound, mammogram and serum tumor markers CEA, CA153 and CA125 (Figure 5F)." -> To which cohort belong these samples? Please indicate in the text and figures.

2. Figure 2: There are several issues related to the analyses and interpretation of the results described in Fig. 2 and Supp. Fig. 2. For example:

o In Fig. 2B, the authors claim an enrichment in gene-body, however, such statement can only be made after comparing the 5mC enrichment in gene-body to the distribution of probes on the whole array (i.e. if the array contains more gene-body-associated probes, it is expected to observe an enrichment in gene-body).

o Fig. 2C is confusing and needs clarifications. In contrast to what has been previously shown, this figure shows a decrease of 5mC in the gene-body. How come? Moreover, are such tiny differences in beta-value even relevant? Is this plot based on all probes or only the differential ones? This panel needs to be better explained, interpreted or suppressed.

o Fig. 2D could be improved by ordering the pathways using some criteria (e.g. significance, score or even similarity between pathways). Additionally, the results should be better discussed: are those pathways really associated with Immuno-surveillance / Immno-editing or do they just reflect the presence of immune cells? Are these pathways really enriched compared to all other pathways? How many immune pathway appear enriched with a random set of genes (control)?

3. The biggest weakness of this study sustains the identification and validation of the final 4 markers:

o The authors state they identified DMPs in PBMC by using a $\Delta\beta \geq 0.06$ and a $p\text{-value} \leq 0.01$. It's unclear if the p-value was corrected for multiple testing. Please clarify and better describe multiple testing in manuscript. Moreover, the authors used an unusual small $\Delta\beta$ to identify DMPs in PBMC. Where does this $\Delta\beta$ come from? Is it meaningful and in accordance with the sensitivity of the used technology? Could the small $\Delta\beta$ be due to the fact that PBMCs consist of different cell types with different epigenomes and a small $\Delta\beta$ may reflect changes in a small subpopulation? If so, the authors should discuss this in the main text and back it up with appropriate references.

o The sub-selection of DMPs lacks clarity. For example:

i. It seems that the authors used a subset of the originally discovered DMPs ($\Delta\beta \geq 0.08$ and $p < 0.0001$) and applied the LASSO machine-learning method to refine it to a set of 6 markers. Regarding this analysis, did the authors use cross-validation? Which criteria were used to select

the signature size (i.e. 6 markers)? None of these key parameters are described in the main text or the Method section.

ii. Why was a $\Delta\beta \geq 0.08$ and $p\text{-value} < 0.0001$ applied before LASSO analysis while it could have been run directly with all the DMPs (except if they re-start from scratch in a cross-validation context???)?

iii. Despite describing in the Methods section a probeLasso method to detect DMR, it seems that this method has not been used anywhere in the paper. Could it be that probeLasso is in fact the LASSO method in the main text? If so, then the DMR associated to identified markers should be mentioned. Also the logic of using DMR as a filter should be explained. Why would a CpG associated to DMR be more likely to become an efficient marker?

iv. It appears odd that two extra markers were added to the final 6 selected by the LASSO analysis. Based on what criteria were they selected? Where they significant in the LASSO analysis? The explanation that "they were located in the 1500 bp of the TSS region and closely related to the BC onset" is very vague and not comprehensible. To keep those two markers, the authors need convincing arguments (e.g. markers have been identified as part of relevant immune pathway in GO analysis). It would also be essential to highlight those two markers in Table 1. Could it be that those two markers are the two last markers in Table 1? As those markers have a $\Delta\beta < 0.08$ they were probably excluded from the LASSO analysis. In such a case, the authors could simply run LASSO using all the DMPs rather than prefilter with $\Delta\beta > 0.08$.

4. PBMCs cell composition: When working with PBMC, an essential parameter to take into consideration is the cell composition of the sample. Are the observed differences in DNA methylation due to a change affecting all cell populations or are they specific to a subpopulation? This question should be investigated using tools that allow for the evaluation of cell populations (e.g. MethyLICIBERSORT).

5. Line 234: "indicating that multiplex analysis was better than any single analysis (Figure 4A)." -> The authors should combine the results of the single analyses into a bioinformatic score to test whether that would improve the performance to the level of the multiplex analysis. When measured in individual assays, the markers could be weighted (weighted score using LASSO parameters) with respect to how much they contribute to the overall diagnostic performance. Indeed, by combining the 4 markers in one PCR assay instead of combining them bioinformatically by generating a score from 4 individual PCR assays greatly limits the possibilities to improve this assay or optimize it when it will be adapted to different clinical scenarios.

6. Figure 6B: In a cohort of healthy samples, using the BC-mqmsPCR assay, the authors predicted 17 positive BC cases among which two were later confirmed as BC. How about the other 15 positive BC cases? Although the short follow-up may partly explain why not more of these cases have been confirmed, it seems unlikely that all of them will develop BC. The authors should better discuss these results and particularly the potential high level of false positive cases of their assay and how this could be handled in the clinic.

Minor comments:

1. Include source that was used for SNP and CR filtering. Minor 1: SNP filtering source is not mentioned. Is it from the ChAMP package or another publication? Also, it is essential to check that the 4 markers do not overlap with cross-reactive probes (see e.g. McCartney et al).

2. Supp. Fig. 2A: PCA plot shows no difference between BC and normal controls. It is therefore misleading to state "to determine whether the methylation profile of BC patients differed from that of normal controls". It is better to describe the PCA as a simple control plot and to include a color code for the different hospitals to assess batch effect.

3. Supp. Fig. 2C: the enrichment mentioned by the author is far from obvious. Is the enrichment significant? And if it is, how can this be interpreted with respect to the overall message of this

paper? Could it be due to CNV bias?

4. Rephrase "...the transcription start sites (TSS) of BC patients were more hypomethylated compared with normal controls" to "...the transcription start sites (TSS) of BC patients were more hypomethylated compared with normal controls".

5. Visually, the delta-beta shown in Fig. 3A seem not to match the ones reported in Table 1. Please clarify and indicate the median in Fig. 3A.

6. The authors should not write "validate the 8 markers" while only 4 markers are actually validating. "Evaluate the reproducibility" would be a better fitting description.

7. Suppl. Fig. 3B: two of the final markers did not validate in TCGA. That is not surprising as TCGA is tumor tissue. This should be better highlighted/explained. In its current form, the TCGA analysis is rather confusing. Also, TCGA analysis needs to be described in Methods section.

8. Fig. 5J: Why is the detection rate of classical methods so low?

9. Fig. 4I&J: Typo in "predicted"

10. Fig. 5F: How was the cutoff defined? Is it always the same cutoff? Describe in main text, not just in Methods section.

11. Fig. 6B: Not clear whether the assessment was made at t0 and/or at follow-up time?

12. Fig. 3: Number of samples should be indicated in box plots.

13. Line 80: "This assay exhibited excellent performance in distinguishing early-stage BC patients, with an AUC of 0.940..." Distinguishing early-stage BC from what?

14. Line 110/111: Change "DNA methylation exists in..." to "DNA methylation changes exist in..."

15. Line 131: Change "...to identify BC-specific methylation landscape" to "...to identify BC-specific methylation landscape in PBMCs"

16. Line 146: "low quality tests" -> Mention/descriptions of these tests is missing in Method section

17. Line 154/155: Change "...identify BC-specific methylation markers" to "...identify BC-specific methylation markers in PBMCs"

18. Line 171: "p-value ≤ 0.01 " is different from the p-value mentioned in methods section for this analysis

19. Line 203: Change "pyrophosphate" to "pyrosequencing"

20. Line 204/205: Change "TCGA database showed hypomethylation levels in BC tissues" to "TCGA database showed hypomethylation in BC tissues"

21. Line 207/208: "...the methylation level of cg18637238 was negatively correlated with tumor size..." -> What about cg16652347?

22. Suppl. Fig. 3C-F: TBS data look better than pyroseq data and could switched (TBS in main and pyroseq in suppl.)

23. Line 231/232: " Δ CT values obtained from BC-mqmsPCR assay were significantly higher than..." -> Provide statistical test

24. Line 363 to 365; "These results suggest that the methylation changes in PBMCs may affect anti-tumor immune surveillance by regulating the expression of immune-related genes, which then impact the occurrence and progression of tumors." -> Statement is too strong and speculative with respect to presented data

25. Line 367 to 369: "our study found that the diagnostic efficacy of DNA methylation markers in PBMCs was higher for early-stage BC than for late-stage BC, and its performance was better for minimal tumors." Incorrect statement as efficacy was compared between early-stage and late-stage BC but instead between early-stage and all-stage BC. Better to say "the diagnostic efficacy of DNA methylation markers in PBMCs was higher in a cohort of early-stage and minimal breast tumors than in a BC cohort of all stages and sizes"

26. Figure 3/4: It would be interesting to see if the 4 final markers (Fig. 3) or the BC-mqmsPCR assay (Fig. 4) behave/perform different when BC cases are subgrouped into Lum A, Lum B, HER2 and TN subtypes. In other words, do the markers or the assay perform better in one BC subtype over others?

27. Figure 4I and 4J: What is the cut-off for positive/negative cases? How was the Cut-off defined? This assay is a key part of the study and should be described in more detail in the main text, not only in the Method section.-> Minor!

Reviewer #3 (Remarks to the Author): expertise in breast cancer detection methods

This is an interesting manuscript. If further confirmatory studies are conducted, it may lead an impact. However, there may be some concerns.

High diagnostic performance seems to be indicated for Stage 0/DCIS. What is the detectability of premalignant lesions, benign breast diseases such as fibroadenoma and fibrocystic disease? Also, as briefly mentioned in the text, is there a difference between low-grade DCIS and high-grade DCIS?

What is the relationship between tumor subtype, grade, and tumor genomic instability/abnormalities? The relationship with pathological biomarkers needs to be summarized more clearly.

The association with tumour stage is indicated. What about that with nodal status, number of nodes involved? If there is a discrepancy in the relationship between tumor size and nodal status, it should be explained.

Is there an association between tumor infiltrating lymphocytes (TILs) and TIL features?

Is there a relationship with profiles, such as the absolute number of lymphocytes in peripheral blood?

What about the performance for invasive lobular carcinoma?

What about time-course/ follow-up data after surgery and also after anti-tumor therapy? Is it possible to show changes of abnormalities that authors focused in this study with respect to the magnitude of abnormalities and also duration of the abnormalities? I would like to know the half-life of this event, if any.

The difference between LC, GC, and CRC is clearly shown, but the mechanism of difference needs to be explained a little more.

Minor comments:

- Inter-institutional sample quality control/assurance should be addressed.
- The data in Figure 6, especially the follow-up data, are very preliminary and maybe too early to

present. It might be better to show it in suppls.

Response to reviewer's comments:

We appreciate the reviewers' positive comments on our manuscript and suggestions on how to improve it further. We have incorporated the suggested edits into the revised manuscript, including updated Figures 1, 2, 3, 5 and 6, as well as Supplemental Figure 3, 6 and the addition of Supplemental Table S3, S4, S6 and S7 to address each of the reviewer's concerns as outlined below. All changes in the text are highlighted in red. Below are our detailed responses (in blue) to each reviewer's specific comments. We hope that these will greatly strengthen the manuscript for publication in Nature Communications.

Reviewer #1

(Remarks to the Author): expertise in machine learning methods

Summary

In this work, Wang et al. introduce a blood-based methylation assay to detect breast cancer from circulating PBMCs. The rationale is early detection of malignancies from non-invasive procedures would be highly beneficial to treatment efficacy and patient outcomes, and builds on existing work in the field using methylation profiles from blood for cancer detection. Here the authors create a well-powered cohort including 820 patients from across 10 hospitals, that are split into three cohorts ("marker discovery", "multistep validation", "assay development and application") that use increasingly fine-grained methylation identification (850k microarray, then pyro/targeted bisulphite sequencing, then a new "BC-mqmsPCR" assay). The BC-mqmsPCR assay is created to simultaneously quantify four methylation markers in a single reaction. The paper identifies four markers that are hypermethylated in breast cancer that validate across the cohorts, and show impressive sensitivity / specificity (92.7%, 85.7%) for early detection of breast cancer. They go on to show that this discriminates from other common cancer types, has higher predictive capacity than existing blood based methylation markers of BC, and may help detect BC cases missed on existing imaging diagnostic systems.

Strengths

- Paper tackles important clinical problem, with state-of-the-art results.
- The dataset used in the study is impressive, with a large sample size and patients from multiple hospitals, which increases the generalizability of the findings.
- The paper is well written and organized, making it easy to follow the research design and results.

Weaknesses

- The biological interpretation of the methylation markers identified is confusing and hard to contextualize with the state-of-the-art biomarker power
- General lack of detail in the computational analyses used to create the diagnostic test
- The train/validation split of the data is mixed across hospitals, which means that an unbiased estimate of the performance of the diagnostic test in a hospital that did not have training data is missing.

Response: We thank the reviewer for the positive evaluation of our work. As for the weaknesses of the manuscript proposed by the reviewer, we revised them point-by-point and gave the replies below.

Major

1. The sample and patient selection in the study raises some questions/requires clarification:

The study states that "low quality tests were removed" but it is unclear what criteria were used to define a "low quality" test.

Response: We thank the reviewer for this suggestion. A "low quality" test refers to an insufficient purity of PBMCs DNA after extraction, $A_{260}/A_{280} \leq 1.6$ or ≥ 2.1 , protein or RNA contamination, and cannot be accurately quantified. We have added it in the revision manuscript (line 479~481).

"Low quality tests (insufficient purity of PBMCs DNA after extraction, such as $A_{260}/A_{280} \leq 1.6$ or ≥ 2.1 , protein or RNA contamination, and cannot be accurately quantified) were removed." (Line 479~481)

The lack of pathological data for some patients raises concerns about whether there could be a correlation between this missing data and the absence of cancer. Can the authors comment on this?

Response: Thank you for this comment. All patients who lacked pathological data were excluded from our study, and all cancer patients we enrolled had pathological information.

The methods section of the paper presents additional information on patient selection criteria (such as no other tumors or autoimmune diseases), but this information should be highlighted earlier in the paper, alongside the sample selection criteria, and discussed as a major limitation of the study. This is because the diagnostic test developed in the study cannot currently be used in patients who have an autoimmune disease or other tumors.

Response: Thank you for this comment. According to the reviewer's suggestion, we have added the sample selection criteria in the result (line 141~147), and discussed their application limitations in the discussion section (line 430~434). As for the evaluation of the diagnostic efficacy of this diagnostic test for other tumors, we included 125 patients with other tumors, including 42 colorectal cancer, 42 lung cancer and 41 gastric cancers, and the results showed that the diagnostic efficacy of the BC-mqmsPCR test for other cancers was not good (AUC: 0.595-0.695).

"The cases were selected by being (i) Pathological diagnosis of BC (without any treatment such as surgery and radiotherapy and chemotherapy); (ii) No other cancers were present; (iii) Subject are no other known inflammatory diseases (bacterial or viral infections, asthma, autoimmune diseases, active thyroid disease) that may alter PBMCs characteristics; (iv) Subjects can understand and sign written informed consent to

participate in the study. Controls were free of malignant diseases and were frequency matched to cases on age and race.” (Line 141~147)

“First, subjects in this study excluded inflammatory diseases that might alter PBMCs characteristics (bacterial or viral infections, asthma, autoimmune diseases, active thyroid diseases), so it is not possible to assess whether the diagnostic tests developed in this study are currently available for patients with inflammatory diseases such as autoimmune diseases.” (Line 430~434)

2. The resulting biomarkers identified raise some questions about what is actually being measured. 3/4 of the markers measured are only really expressed in T/NK cells (human protein atlas). Consequently, it's really hard to understand how this test measures anything other than T/NK cell burden in the blood since KLR genes aren't specific to breast cancer, but NK cells in general. This interpretation is somewhat backed up by the fact that the authors find the opposite pattern of methylation in TCGA (vs normal tissue) and that the discriminatory power of the test decreases as tumour size increases (you would somewhat expect the opposite as a larger tumour would shed more / illicit more of an immune response). While this doesn't invalidate the use of this probe as a biomarker, it makes understanding what's actually happening really difficult. I would be interested in the other reviewers' thoughts in this matter, and would encourage the authors to look at the predictive power of their biomarkers in an entirely independent cohort (e.g. Tang et al. Oncotarget 2016 as referenced in their paper).

Response: We appreciate the reviewer's insightful and constructive comments. Cancer is a systemic disease that induces many functional and compositional changes to the immune system as a whole¹. It has been proven that methylation changes of peripheral blood mononuclear cells are associated with cancer²⁻⁴, so measuring the methylation changes of peripheral blood mononuclear cells may be able to predict cancer. Natural killer cells have been initially identified as a lymphoid population representing the 10–20% of PBMC. As the first line of defense of human health, natural killer cells can be quickly activated when tumor cells are detected, directly killing cancer cells, while secreting "chemokines" and recruiting dendritic cells. The activated dendritic cells then carry tumor antigens to the lymph nodes, presenting the signature of the cancer to "killer" T cells, which then travel to battle with NK cells to kill the cancer⁵.

Our study is to explore the role of DNA methylation markers of peripheral blood mononuclear cells in the early screening of breast cancer. The four sites finally screened are immune cell related receptors, which is consistent with the main body of our study, reflecting the epigenetic regulation of the host immune system in the occurrence and development of cancer. As the reviewers noted, KLRK1 is not specific to breast cancer, which may raise the question of whether other cancers cause corresponding host changes. To confirm the diagnostic efficacy of these gene methylation markers in other cancers, we included 125 patients with other cancers, and the results showed that these methylation markers were less effective in the diagnosis of other cancers (**Figure 4K-L below**). We also conducted a peripheral blood mononuclear cell methylation marker study in lung, colorectal, and gastric cancers. The results showed that the methylation changes of peripheral blood mononuclear cells caused by different cancers were

specific. As for the unconventional phenomenon that the degree of methylation decreases with the increase of tumor size, we have observed this phenomenon in the methylation sequencing results of peripheral blood mononuclear cells including breast cancer, gastric cancer and colorectal cancer (unpublished data). In addition, Zhang et al. also showed similar results to ours in their study on DNA methylation markers of peripheral blood mononuclear cells of liver cancer⁴. We speculate that this is because the immune system is activated and in a dominant position in the early stage of tumor development, and gradually suppressed as the tumor progresses, so the degree of methylation decreases. We demonstrated the predictive power of these four methylation markers using pyrosequencing and targeted bisulfite sequencing in a separate cohort II, as shown in **Figure 3D** and **Supplementary Figures 3D** below.

Figure 4. (K) Distribution of BC-mqmsPCR score in BC (n=206), LC (n=42), GC (n=41), CRC (n=42) and normal controls (n=170). The red dashed line represents the cut-off value (3.223). LC, lung cancer; GC, gastric cancer; CRC, colorectal cancer. (L) The AUC for distinguishing BC from non-BC was 0.913.

Figure 3D and Supplementary Figures 3D. ROC curves of 4 methylation markers in all BC, stage 0/I BC and ≤1.5 cm BC of the pyrosequencing and targeted bisulfite sequencing set.

3. The computational analysis performed requires additional details:

Normalization strategy - SVA and combat are used for normalization, does this happen across the train / validation cohort simultaneously? If so this would count as train/test leakage. Similarly, what (if any) normalization procedure is used for the BC-mqmsPCR assay?

Response: Thank the reviewer for this suggestion. Normalization strategy - SVA and combat are not used in the train/validation cohort. For the detection of BC-mqmsPCR assay, we used single internal control gene for normalization, using ACTB as the internal reference gene, Δ CT value between the target molecule and the internal reference gene were used to indicate methylation level, and this method was also used by Jin et al⁶.

GO and Kegg enrichment performed “using R package”. Which R package? What parameters?

Response: Thank the reviewer for this suggestion. GO and Kegg enrichment performed “using R 4.0.0 package”. To clarify the biological functions of the genes and the involved signaling pathways, we annotated each Gene based on the Gene Ontology and KEGG database. Enrichment calculations were performed using Fisher's exact test. Further, we also need to conduct GO and pathway enrichment analysis of the target genes. The specific principle is to carry out annotation mapping of differentially expressed genes in GO and KEGG database entries, calculate the number of the target genes in each GO and pathway entry, and then use hypergeometric test for statistics. Select the GO and KEGG entries that are significantly enriched in the differentially expressed genes. After the calculated p-value was corrected by multiple hypothesis tests, the P-value 0.05 was taken as the threshold, and the GO and KEGG term meeting this condition was defined as the GO and KEGG term significantly enriched in the target genes,

Rich Factor, whose calculation formula is:

$$\left(\frac{\text{diff_gene_in_this_pathway}}{\text{diff_gene_in_all_pathway}} \right)$$

$$\left(\frac{\text{all_gene_in_this_pathway}}{\text{all_gene_in_all_pathway}} \right)$$
 . It was drawn by ggplot2.

The figure shows the results of ranking the top 30 items in terms of enrichment degree (Rich Factor).

It's written for pyrosequencing “Absolute methylation levels of CpG sites were calculated by software”. Which software and how are they calculated?

Response: Thank the reviewer for this suggestion. The Pyro Q-CPG software of the pyrosequencing device was used to automatically analyze the methylation status of each site. Analysis of the difference between the target sequence and the actual read sequence information

4. The initial set of methylation sites were chosen via various thresholds (e.g. $p < 0.0001$), but this only returns 6 so an extra 2 are added “because they were located in the 1500bp of the TSS region and closely related to the BC onset”. This requires more details. What is the rationale for ad-hoc adding in extra genes? TSS region for which

gene and how were these genes chosen? How was this chosen? What paper shows a link to BC onset? Were these 2 genes part of the final 4 that are hypermethylated and validate, or hypomethylated and don't validate?

Response: We apologize for the confusion. As shown in **Table 1 below**, according to screening criteria (e.g. $p < 0.0001$), 6 methylation sites were screened (the blue part). It can be seen that most methylation sites are located in the Body region, and only cg11754974 is located in TSS1500. Given the notion that methylation in the TSS region has been shown to be more important than other CpG sites in regulating gene expression⁷, we included 2 more methylation sites in the TSS region that might be associated with the development of breast cancer (highlighted in red, cg13828440/KLRD1 and cg18637238/KLRK1).

Varker et al. showed that cancer-related stress in patients with invasive breast cancer is related to immune impairment. In patients exhibiting high levels of cancer-related psychological stress, NK cytotoxicity is reduced and CD94 (KLRD1) levels are reduced⁸. Starčević et al. confirmed that the regulation of NK cell activation in tumor invasion may be involved in the pathogenesis of breast cancer. The expression of peripheral NK cell-activated receptors (CD94/NKG2C, NKG2D, and CD16) and inhibitory receptors (NKG2A) from breast cancer patients was reduced, highlighting the importance of NK cells as suitable targets for effective antitumor responses in the breast cancer immunosuppressive tumor microenvironment⁹.

Kruijf et al. confirmed that NKG2D ligand (NKG2DL) is often highly expressed in breast cancer, and NKG2DL can bind to the activating NKG2D (KLRK1) receptor present on NK cells and subsets of T cells, thus initiating immune response and playing a crucial role in tumor immune editing in breast cancer¹⁰. Other studies have linked NKG2D to the development of breast cancer¹¹⁻¹³. These 2 genes were part of the final 4 that are hypermethylated and validate (cg11754974/TRDJ3, cg16652347/PLXNA4, cg13828440/KLRD1 and cg18637238/KLRK1). We explain this in the revised manuscript (Line 194~201).

“Next, we applied a series of screening principles to identify the most important and specific DNA methylation markers of BC. By using LASSO analysis and $|\Delta\beta| \geq 0.08$, $p\text{-value} \leq 0.0001$ as the screening criteria, we obtained 6 methylation sites (Top 6 methylation sites in Table 1). It can be seen that most methylation sites are located in the Body region, and only cg11754974 is located in TSS1500. Given the notion that methylation in the TSS region has been shown to be more important than other CpG sites in regulating gene expression (31), we included 2 more methylation sites (the bottom 2 methylation sites in Table 1) in the TSS region that might be associated with the development of BC (32-37).” (Line 194~201)

31. Hong J & Rhee JK (2022) Genomic Effect of DNA Methylation on Gene Expression in Colorectal Cancer. *Biology (Basel)* 11(10).
32. Starcevic A, et al. (2022) Differences in tolerogenic status of NK cells between luminal A type, luminal B type, and triple-negative breast cancer. *Neoplasma* 69(6):1289-1302.
33. de Kruijf EM, et al. (2012) NKG2D ligand tumor expression and association

with clinical outcome in early breast cancer patients: an observational study. *BMC Cancer* 12:24.

34. Varker KA, *et al.* (2007) Impaired natural killer cell lysis in breast cancer patients with high levels of psychological stress is associated with altered expression of killer immunoglobulin-like receptors. *J Surg Res* 139(1):36-44.
35. Mamessier E, *et al.* (2011) Human breast tumor cells induce self-tolerance mechanisms to avoid NKG2D-mediated and DNAM-mediated NK cell recognition. *Cancer Res* 71(21):6621-6632.
36. Raab S, *et al.* (2014) Fc-optimized NKG2D-Fc constructs induce NK cell antibody-dependent cellular cytotoxicity against breast cancer cells independently of HER2/neu expression status. *J Immunol* 193(8):4261-4272.
37. Verma C, *et al.* (2015) Natural killer (NK) cell profiles in blood and tumour in women with large and locally advanced breast cancer (LLABC) and their contribution to a pathological complete response (PCR) in the tumour following neoadjuvant chemotherapy (NAC): differential restoration of blood profiles by NAC and surgery. *J Transl Med* 13:180.

Table 1. Features of 8 differentially methylated CpG positions (DMPs) among breast cancer (BC) and normal controls (NC) group.

probeID	$\Delta\beta$ value (BC-NC)	P.Value	adjust.P.Value	CHR	UCSC_RefGene	UCSC_RefGene_Group
cg14507403	-0.151771106	4.92E-05	0.047018976	7	LMTK2	Body
cg09821790	-0.09869485	5.12E-05	0.047837259	16	SLC7A6	Body
cg15694422	-0.094113112	2.92E-07	0.01187436	3	MGLL	Body
cg27527887	-0.086760967	6.52E-05	0.05261645	10	/	/
cg11754974	0.081256084	9.46E-05	0.057515784	14	TRDJ3	TSS1500
cg16652347	0.084377724	2.97E-05	0.039173184	7	PLXNA4	Body
cg13828440	0.068921275	4.34E-06	0.021774546	12	KLRD1	TSS1500
cg18637238	0.061870615	2.68E-05	0.037570473	12	KLRK1	TSS1500

probe ID: identification of the probe (from 850K BeadChip Array). CHR: chromosome. TSS1500: 1500 bp of the start site of transcription.

5. It is not sufficiently detailed what training actually happens in the training/validation step for BC-mqms stage:

Is this to find a threshold that maximizes the Youden index which is then applied to the validation set?

Response: Yes, we find a threshold that maximizes the Youden index in the training set and then apply it to the validation set, which we have explained in the revised manuscript (Line 571~572).

“Positive and negative classifications of BC-mqmsPCR were determined by the cutoff value (3.223) using Youden’s index in the training set.” (Line 571~572)

How are the four Δ CT variables (for each of the four selected CpG sites) combined into a single score?

Response: We apologize for the confusion. Our primary objective is to develop a simple multiple quantitative methylation-specific PCR method for simultaneous quantification of multiple DNA methylation markers. Thus, we designed BC-mqmsPCR assay with two different fluorophore probes. The FAM fluorophore was used for four selected CpG sites, and the VIC fluorophore was used for the ACTB control assay. This BC-mqmsPCR assay measures the total methylation of four selected CpG sites. The signals collected by FAM fluorescence groups represent the total methylation of the four selected CpG sites. The Δ CT value represented the methylation level (Δ CT = CT reference – CT biomarker), and only one Δ CT value was obtained by BC-mqmsPCR assay.

Figure 4K presents a “BC-mqmsPCR Probability score”. This is not referenced anywhere else (and doesn’t range between 0 and 1 like a probability should).

Response: We apologize for the confusion. BC-mqmsPCR Probability score is actually the Δ CT value of each sample, so it is not between 0 and 1. We have explained this in the revised manuscript (Line 836-838).

“The red dashed line represents the cut-off value (3.223), BC-mqmsPCR Probability score is the Δ CT value of each sample.” (Line 836-838)

6. The training and validation cohorts mix across hospitals. In other words, the AUC scores presented are equivalent to the performance expected if there was always training data collected from the hospital in which the test will be deployed. Can the authors perform an analysis where training/validation are stratified across hospitals to quantify the performance expected if training data was not available from a given hospital?

Response: We apologize for the confusion caused to the reviewers. In fact, our training cohort and validation cohort were not mixed across hospitals. We included training set samples from six centers for methodology development and search for the optimal cut-off value, and then included validation set samples from another four centers for external verification. The results showed that our detection method had scalability and was stable and robust in clinical application. We have explained this in the revised manuscript (Line 263~268).

“To further assess the performance of BC-mqmsPCR assay for clinical application, we recruited a multicenter cohort of 206 BC patients and 170 normal controls from 10 hospitals in China and divided into training and validation sets. Training set (130 BC + 115 normal controls) were recruited from 6 centers for methodology development and search for the optimal cut-off value, and validation set (76 BC + 55 normal controls)

were recruited from another 4 centers for external validation.” (Line 263~268)

7. On line 383 authors write that other methylation biomarkers have been proposed for early BC detection (e.g. RPTOR) and state some AUCs on cohorts. Can the authors clarify if the AUCs / cohorts listed are their own or the original papers’ results? If original papers, can they quantify the predictive power of these markers in their data?

Response: The AUCs / cohorts listed in line 383 is the result of the original paper. In the original paper by Tang et al., the authors only quantified the combined diagnostic efficacy of three markers (RPTOR, MGRN1 and RAPSN) in cohorts I, II and III, and did not quantify the diagnostic efficacy of a single marker. The paragraphs in red subscript are descriptions of relevant results in the original paper.

“To estimate the potential power of these three genes with regard to differentiate the BC cases from the controls, ROC curve analysis was performed by logistic regression using the backwards conditional variable selection method and adjusting for age and experimental batches. The model was built on the data set of validation cohort I, revealing an internal AUC of 0.79 (95%CI 0.73-0.85) and validated externally in both validation cohorts II and III with AUCs of 0.60 (95%CI 0.54-0.66) and 0.62 (95%CI 0.57-0.67), respectively (Figure 2).”

Figure 2: The diagnostic potential of the combined marker panel (RPTOR, MGRN1 and RAPSN) for differentiating breast cancer cases from healthy controls. ROC curves for logistic regression models based on the combination of RPTOR, MGRN1 and RAPSN in three validation rounds. Backwards conditional variables selection method was used in the logistic regression

Minor

- Figure 1 typo “pathology”

Response: We apologize for this typo and have corrected it in the revised manuscript (Revised Figure 1).

Revised Figure 1. Workflow chart of the study design.

- Missing citation(s) for “DNA methylation exists in almost all cancers and occurs in precancerous or early cancer stage”

Response: We appreciate the reviewer's suggestions and have added the appropriate references to the revised manuscript (References 11-14).

11. Roy D & Tiirikainen M (2020) Diagnostic Power of DNA Methylation Classifiers for Early Detection of Cancer. *Trends Cancer* 6(2):78-81.
12. Qiao G, *et al.* (2021) Discovery and validation of methylation signatures in circulating cell-free DNA for early detection of esophageal cancer: a case-control study. *BMC Med* 19(1):243.
13. Liang W, *et al.* (2019) Non-invasive diagnosis of early-stage lung cancer using high-throughput targeted DNA methylation sequencing of circulating tumor DNA (ctDNA). *Theranostics* 9(7):2056-2070.
14. Laird PW (2003) The power and the promise of DNA methylation markers. *Nat Rev Cancer* 3(4):253-266.

Reviewer #2

(Remarks to the Author): expertise in DNA methylation in breast cancer

In this study, Wang et al. used Infinium 850K BeadChips to identify changes in DNA methylation in peripheral blood mononuclear cells (PBMCs) between patients with breast cancer (BC) and healthy controls. Using pyrosequencing and targeted bisulphite sequencing, Wang et al. validated four of the BC-specific PBMCs markers and developed, based on the four markers, a multiplex methylation-specific quantitative PCR assay for the detection of BC. Wang et al. validated the performance of this assay in a multicenter cohort and highlighted its value for the diagnosis of early-stage BC. This study is very interesting and timely. While the validation of the performance of the BC-mqmsPCR assay appears convincing, the identification and validation of the BC-specific PBMCs markers is compromised by several major weaknesses.

Response: We appreciate the reviewer's positive comments. We have made a large number of revisions to the identification and validation of BC-specific PBMCs markers, which have several major weaknesses. We hope that the revised manuscript can solve the above problems.

Major comments:

1. The description of the cohorts, especially the subsampling of the different cohorts throughout the paper, is confusing and should be clarified. For example:

Line 236-237: "...qMSP assay to detect the methylation levels of 40 BC patients and 40 normal controls." -> To which cohort belong these samples? Please indicate in the text.

Response: We apologize for the confusion. These samples belong to the training set in Cohort III. We have indicated the source of the samples in the revised manuscript (Line 246~249).

"To validate whether BC-mqmsPCR Assay could be used to distinguish BC patients from normal controls, we performed 4-marker BC-mqmsPCR assay and individual marker qMSP assay to detect the methylation levels of 40 BC patients and 40 normal controls (Derived from the training set in Cohort III)." (Line 246~249).

Line 269-270: "...the methylation level was significantly higher in stage 0/I and II BC patients than stage III patients..." -> Where do these samples come from and how many where there for each stage? Please clarify the cohort as well as the sample number per stage in text and figures (Fig. 1 & Fig. 5A)

Response: These samples are come from the training and validation sets of Cohort III. A total of 245 patients were enrolled in the training set, including 115 normal patients and 130 patients with breast cancer, including 59 patients in stage 0-I, 56 patients in stage II, 10 patients in stage III-IV, and 5 patients unclassified. A total of 131 patients were enrolled in the verification set, including 55 normal patients and 76 patients with breast cancer, including 28 patients in stage 0-I, 30 patients in stage II, 12 patients in stage III-IV, and 6 patients not classified. We have illustrated the cohort as well as the

sample number per stage in **Revised Figures 1** and **Revised Figure 5A** in the revised manuscript.

Revised Figures 1: Workflow chart of the study design.

Revised Figures 5: The application of BC-mqmsPCR in the detection of minimal tumor and early-stage BC.

Line 296-297: "...in a cohort of 67 patients with early-stage BC and 113 patients with small to medium-sized mass BC..." -> Where do these samples come from and how many where there for each stage? Please clarify the cohort as well as the sample number

per stage in text and figures (Fig. 1 & Fig. 5J)

Response: The cohort of 67 patients with early-stage BC were come from stage 0-1 BC patients with complete CEA, CA153, CA125 and BC-mqmsPCR information in the training set and validation set of Cohort III, including 44 patients in the training set and 23 patients in the validation set. The cohort of 113 patients with small to medium-sized mass BC were come from tumor size ≤ 2.5 cm BC patients with complete CEA, CA153, CA125 and BC-mqmsPCR information in the training set and validation set of Cohort III, including 71 patients in the training set and 42 patients in the validation set. We have illustrated the cohort as well as the sample number per stage in text (line 311~316) and **Revised Figures 1** in the revised manuscript.

“We compared the performance between BC-mqmsPCR and currently used tumor markers CA153, CEA and CA125 in a cohort of 67 patients with early-stage BC (stage 0-1 BC patients with complete CEA, CA153, CA125 and BC-mqmsPCR information in the training set and validation set) and 113 patients with small to medium-sized mass BC (tumor size ≤ 2.5 cm BC patients with complete CEA, CA153, CA125 and BC-mqmsPCR information in the training set and validation set), respectively.” (Line 311~316)

Line 316-317: “... follow-up study on 170 healthy control patients in the training and validation sets...” -> To which cohort belong these samples? Please indicate in the text.

Response: The 170 healthy control patients were from all healthy patients in the training set and validation set of Cohort III, including 115 patients in the training set and 55 patients in the validation set. We have indicated the source of the sample in line 334~336 in the revised manuscript.

“Therefore, we conducted a follow-up study on 170 normal control patients in the training (115 normal controls) and validation sets (55 normal controls)” (Line 334~336)

Line 278-280: “...two cases that were diagnosed by BC-mqmsPCR but missed by ultrasound, mammogram and serum tumor markers CEA, CA153 and CA125 (Figure 5F).” -> To which cohort belong these samples? Please indicate in the text and figures.

Response: These two samples are from training set of Cohort III. We explain the source of samples in line 294~296 of the revised manuscript.

“The potential utility of this approach was highlighted in two cases (from training set of Cohort III) that were diagnosed by BC-mqmsPCR but missed by ultrasound, mammogram and serum tumor markers CEA, CA153 and CA125 (Figure 5F).” (Line 294~296).

2. Figure 2: There are several issues related to the analyses and interpretation of the results described in Fig. 2 and Supp. Fig. 2. For example: In Fig. 2B, the authors claim an enrichment in gene-body, however, such statement can only be made after comparing the 5mC enrichment in gene-body to the distribution of

probes on the whole array (i.e. if the array contains more gene-body-associated probes, it is expected to observe an enrichment in gene-body).

Response: Agreed. This claim is not accurate enough and we have modified the description in line 180~181 of the revised manuscript.

“The genomic distribution of DMPs showed that hypo-DMPs and hype-DMPs accounted for 31.6% and 23.0% of the gene body region, respectively.” (Line 180~181).

Fig. 2C is confusing and needs clarifications. In contrast to what has been previously shown, this figure shows a decrease of 5mC in the gene-body. How come? Moreover, are such tiny differences in beta-value even relevant? Is this plot based on all probes or only the differential ones? This panel needs to be better explained, interpreted or suppressed.

Response: We apologize for the confusion. Figure 2C describes the inter-group methylation differences between the breast cancer group and the healthy control group at the DMPs distributed at 2kb upstream and downstream of the TSS. It is based on differential methylation sites. In order not to confuse the readers, we have changed the color of the group in **Revised Figures 2C** of revised manuscript.

Revised Figures 2C. Distribution of upstream and downstream methylation of TSS in PBMCs of BC patients and normal controls. TSS, transcription start site.

Fig. 2D could be improved by ordering the pathways using some criteria (e.g. significance, score or even similarity between pathways). Additionally, the results should be better discussed: are those pathways really associated with Immuno-surveillance / Immno-editing or do they just reflect the presence of immune cells? Are these pathways really enriched compared to all other pathways? How many immune pathways appear enriched with a random set of genes (control)?

Response: Thanks for the reviewer's suggestion. We improved Figure 2D by ranking the pathways by significance (q value). The significance gradually increased from top to bottom. To clarify the biological functions of the genes and the involved signaling pathways, we annotated each Gene based on the Gene Ontology. Enrichment calculations were performed using Fisher's exact test. Further, we conduct GO enrichment analysis of the target genes. The specific principle is to carry out annotation mapping of differentially expressed genes in GO entries, calculate the number of the target genes in each GO entry, and then use hypergeometric test for statistics. Select the GO entries that are significantly enriched in the differentially expressed genes. After the calculated p-value was corrected by multiple hypothesis tests, the P-value 0.05 was taken as the threshold, and the GO term meeting this condition was defined as the GO term significantly enriched in the target genes.

The top-ranking enriched GO terms were involved in immune functions¹⁴⁻¹⁶, such as regulation of B cell proliferation, regulation of myeloid leukocyte differentiation, and regulation of lymphocyte apoptotic process, supporting the hypothesis that DNA methylation changes in PBMCs of BC patients are associated with host immune responses. For example, apoptosis in T and B lymphocytes is involved in all fundamental processes in the immune system. It is a mechanism to regulate the course of an immune response and to establish immunological memory as well as central and peripheral tolerance. New findings in the regulation of apoptosis in normal T and B lymphocytes have had a major impact on the understanding of central and peripheral tolerance, the regulation of an immune response and the development of immunological memory. This understanding will also be key to the elucidation of the pathogenesis of autoimmune diseases, T-cell depletion in AIDS, and malignant growth.

We order the pathways by significance in **Revised Figures 2D** and add a discussion of this result in line 383-387 of the revised manuscript.

“The differentially methylated genes were enriched in immune functions (51-53), such as regulation of B cell proliferation, regulation of myeloid leukocyte differentiation, and regulation of lymphocyte apoptotic process, supporting the hypothesis that DNA methylation changes in PBMCs of BC patients are associated with host immune responses.” (Line 383-387)

D
Revised Figures 2D. The GO enrichment results of differentially methylated genes showed that multiple items were involved in the immune system and immune surveillance system.

3. The biggest weakness of this study sustains the identification and validation of the final 4 markers:

①The authors state they identified DMPs in PBMC by using a $\Delta\beta \geq 0.06$ and a $p\text{-value} \leq 0.01$. It's unclear if the $p\text{-value}$ was corrected for multiple testing. Please clarify and better describe multiple testing in manuscript. Moreover, the authors used an unusual small $\Delta\beta$ to identify DMPs in PBMC. Where does this $\Delta\beta$ come from? Is it meaningful and in accordance with the sensitivity of the used technology? Could the small $\Delta\beta$ be due to the fact that PBMCs consist of different cell types with different epigenomes and a small $\Delta\beta$ may reflect changes in a small subpopulation? If so, the authors should discuss this in the main text and back it up with appropriate references.

Response: We thank the reviewer for the helpful suggestions. The p value used to define DMPs is the p value that has not been corrected for multiple testing, because after using the Benjamini & Hochberg method to correct the p value, we found that there are only 8 methylation sites with $\text{adjust } p\text{-value} \leq 0.01$, and the number is too small. In view of the small difference of methylation in PBMCs samples between breast cancer and healthy control group, we used $\Delta\beta \geq 0.06$ to define DMPs. In the study of DNA methylation markers based on peripheral blood or PBMCs sources, other

researchers also chose a smaller delta-beta value. For example, Tang et al. use $|\Delta\beta| > 0.04$ as threshold to find the difference in peripheral blood DNA methylation between sporadic breast cancer cases and control groups¹⁷. Zhu et al. use $|\Delta\beta| > 0.05$ as threshold for differentiating rheumatoid arthritis patients and health controls¹⁸. Chaix et al. use $|\Delta\beta| > 0.03$ as distinguishing DNA methylation differences in experienced meditators after an intensive day of mindfulness-based practice¹⁹. Therefore, we consider it acceptable to set the $\Delta\beta$ value for identifying DMPs to 0.06. We have clarified this in line 504~505 and added references in line 174~175 of the revised manuscript.

“The CpG positions with $|\Delta\beta| \geq 0.06$ and p-value ≤ 0.01 were defined as the differentially methylated CpG positions (DMPs) (26-28)” (Line 174~175)

26. Zhu H, *et al.* (2019) Rheumatoid arthritis-associated DNA methylation sites in peripheral blood mononuclear cells. *Ann Rheum Dis* 78(1):36-42.
27. Tang Q, *et al.* (2016) DNA methylation array analysis identifies breast cancer associated RPTOR, MGRN1 and RAPSN hypomethylation in peripheral blood DNA. *Oncotarget* 7(39):64191-64202.
28. Chaix R, *et al.* (2020) Differential DNA methylation in experienced meditators after an intensive day of mindfulness-based practice: Implications for immune-related pathways. *Brain Behav Immun* 84:36-44.

“CpGs having $|\Delta\beta| \geq 0.06$ and p-value ≤ 0.01 were considered as DMPs (p-value has not been corrected by multiple tests)” (Line 504~505)

As to whether the small delta-beta might be due to the fact that PBMCs consist of different cell types with different epigenomes and a small delta-beta may reflect changes in a small subpopulation, we analyzed each cellular component in PBMCs using EpiDISH method²⁰ published in *Nature methods*. we found significantly higher monocytes proportion in BC patients than in controls, and NK cells proportion in BC were significantly lower than those in normal controls, but not in CD8 + T cells, CD4 + T cells, B cells and granulocytes (**Figure 1 below and Revised Table S3 below**). Nevertheless, adjusted for the cell proportions, the eight CG sites we screened still showed significant difference between BC cases and controls (**Revised Table S4 below**). We have added the following statement as well as Revised Tables S3 and S4 to the results in the revised manuscript (Line 203~209).

“To rule out the influence of cell mixture distribution in PBMCs on DNA methylation, we calculated the cell proportion according to EpiDISH method (38), and found significantly higher monocytes proportion in BC patients than in controls, and NK cells proportion in BC were significantly lower than those in normal controls, but not in CD8 + T cells, CD4 + T cells, B cells and granulocytes (**Revised Table S3**). Nevertheless, adjusted for the cell proportions, the 8 CG sites we screened still showed significant difference between BC cases and controls (**Revised Table S4**).” (Line 203~209)

38. Zheng SC, Breeze CE, Beck S, & Teschendorff AE (2018) Identification of differentially methylated cell types in epigenome-wide association studies. *Nat Methods* 15(12):1059-1066.

Figure.1 The box plots present the distribution of cellular components in BC patients and normal controls. The X-axis represents each cell fraction. BC: breast cancer

Revised Table S3 Summary table of results of two-way ANOVA based on cell components

ID	Pr(>F) Group	Fvalues_Group	levene.Fvalue	levene.pvalue	shapino.Wvalue	shapino.pvalue	mean_BC	mean_Normal	sd_BC	sd_Normal
NK cells	0.000685938	12.50715707	1.532939929	0.219385399	0.979227545	0.218495904	0.233244137	0.307490116	0.082656365	0.103360388
Monocytes	0.005408004	8.188995686	3.315194167	0.072477333	0.94984011	0.003375059	0.185993509	0.150786642	0.056042147	0.048236214
Granulocytes	0.205399564	1.630664497	5.365943508	0.023160218	0.955310587	0.007026589	0.104854039	0.083931471	0.078846761	0.055081803
CD4+T cells	0.330760925	0.957826294	0.004804066	0.944919159	0.979370786	0.222993736	0.239582505	0.221635337	0.080228883	0.077995816
B cells	0.477784831	0.508806313	0.192740084	0.661859558	0.974276856	0.106444798	0.088582415	0.083257673	0.033174081	0.030833999
CD8+T cells	0.750400075	0.101912367	0.172116621	0.679376396	0.981844676	0.314824374	0.147743395	0.152898762	0.071118069	0.067867854

Revised Table S4 Table of EWAS results based on Logistic regression models.

probeID	Estimate	StdError	Zvalue	Pvalue
cg11754974	0.176546375	0.07317412	2.412688722	0.015835337
cg16652347	0.225803649	0.067394859	3.350458092	0.00080678
cg13828440	0.261032927	0.114833316	2.273146299	0.023017365
cg18637238	0.238724796	0.10023098	2.381746598	0.01723075
cg14507403	-0.060169757	0.018088697	-3.32637322	0.00087984
cg09821790	-0.061352485	0.030573082	-2.006748455	0.044776454
cg15694422	-0.257659442	0.070673533	-3.645769942	0.000266592
cg27527887	-0.141380207	0.045976006	-3.075086728	0.002104413

The sub-selection of DMPs lacks clarity. For example:

i. It seems that the authors used a subset of the originally discovered DMPs ($\Delta\beta \geq 0.08$ and $p < 0.0001$) and applied the LASSO machine-learning method to refine it to a set of 6 markers. Regarding this analysis, did the authors use cross-validation? Which criteria were used to select the signature size (i.e. 6 markers)? None of these key parameters are described in the main text or the Method section.

Response: We are very sorry that we did not provide a detailed explanation of the selection criteria for candidate markers in the article. We referred to the screening principles of DNA methylation candidate markers from different researchers^{17, 21}, and then based on the sequencing data of the 850k chip in this study, choose to use a LASSO and $|\Delta\beta| \geq 0.08$, $p\text{-value} \leq 0.0001$ two sets of screening criteria to further narrow the selection range of candidate markers. Using the LASSO method (10-fold cross-validation), 33 methylation markers were obtained (**Revised Table S6**), and using $|\Delta\beta| \geq 0.08$, $p\text{-value} \leq 0.0001$ screening principle, 8 methylation markers were obtained (**Revised Table S7**), with 6 methylation markers overlapped in the two screening criteria. After removing 2 methylation markers (cg21723696 and cg14928964) that were difficult to design with primers, we ultimately selected 4 overlapping methylation markers (cg14507403, cg09821790, cg15694422 and cg27527887) and 2 methylation markers (cg11754974 and cg16652347) that only met $|\Delta\beta| \geq 0.08$, $p\text{-value} \leq 0.0001$ screening principle for further validation. We have added the screening criteria for candidate markers in the methods section of the revised manuscript (Line 508~521).

“Selection of candidate methylation markers: Candidate methylation markers were selected according to LASSO, $\Delta\beta$, $p\text{-value}$, and location. Using the LASSO method (10-fold cross-validation), 33 methylation markers were obtained (Table S6), and using $|\Delta\beta| \geq 0.08$, $p\text{-value} \leq 0.0001$ screening principle, 8 methylation markers were obtained (Table S7), with 6 methylation markers overlapped in the two screening criteria. After removing 2 methylation markers (cg21723696 and cg14928964) that were difficult to design with primers, we ultimately selected 4 overlapping methylation markers (cg14507403, cg09821790, cg15694422 and cg27527887) and 2 methylation markers (cg11754974 and cg16652347) that only met $|\Delta\beta| \geq 0.08$, $p\text{-value} \leq 0.0001$ screening

principle for further validation. Given the notion that methylation in the TSS region has been shown to be more important than other CpG sites in regulating gene expression, we included 2 more methylation sites (cg13828440 and cg18637238) in the TSS region that might be associated with the development of BC. Finally, 8 candidate markers were selected for further validation.” (Line 508~521)

ii. Why was a $\Delta\beta \geq 0.08$ and $p\text{-value} < 0.0001$ applied before LASSO analysis while it could have been run directly with all the DMPs (except if they re-start from scratch in a cross-validation context???)?

Response: We apologize for the confusion caused to the reviewers. The screening principles of $|\Delta\beta| \geq 0.08$, $p\text{-value} \leq 0.0001$ and LASSO analysis operate independently, as we explained in the previous question.

iii. Despite describing in the Methods section a probeLasso method to detect DMR, it seems that this method has not been used anywhere in the paper. Could it be that probeLasso is in fact the LASSO method in the main text? If so, then the DMR associated to identified markers should be mentioned. Also the logic of using DMR as a filter should be explained. Why would a CpG associated to DMR be more likely to become an efficient marker?

Response: We apologize for the confusion. ProbeLasso method is not LASSO method. We used ProbeLasso to calculate DMR, but did not show the relevant results of DMR in the paper. Instead, we chose DMP for marker study. Therefore, in general consideration, we deleted the description of ProbeLasso in the method and added the description of LASSO in the revised manuscript (Line 509~511).

“Candidate methylation markers were selected according to LASSO, $\Delta\beta$, $p\text{-value}$, and location. Using the LASSO method (10-fold cross-validation), 33 methylation markers were obtained.” (Line 509~511)

iv. It appears odd that two extra markers were added to the final 6 selected by the LASSO analysis. Based on what criteria were they selected? Where they significant in the LASSO analysis? The explanation that “they were located in the 1500 bp of the TSS region and closely related to the BC onset” is very vague and not comprehensible. To keep those two markers, the authors need convincing arguments (e.g. markers have been identified as part of relevant immune pathway in GO analysis). It would also be essential to highlight those two markers in Table 1. Could it be that those two markers are the two last markers in Table 1? As those markers have a $\Delta\beta < 0.08$ they were probably excluded from the LASSO analysis. In such a case, the authors could simply run LASSO using all the DMPs rather than prefilter with $\Delta\beta > 0.08$.

Response: Thanks for the reviewer's suggestion. Yes, these two markers are the two last markers in Table 1, and we have highlighted them in the revised manuscript. We chose these two markers because we found that most of the 6 markers screened by criteria (LASSO analysis and $|\Delta\beta| \geq 0.08$, $p\text{-value} \leq 0.0001$) were located in the body region, while only cg11754974 were located in the TSS region. Considering that

methylation in the TSS region has been shown to be more important than other CpG sites in regulating gene expression⁷, we included two additional methylation sites in the TSS region that may be involved in breast cancer development (cg13828440/KLRD1 and cg18637238/KLRK1). Varker et al. showed that cancer-related stress in patients with invasive breast cancer is related to immune impairment. In patients exhibiting high levels of cancer-related psychological stress, NK cytotoxicity is reduced and CD94 (KLRD1) levels are reduced⁸. Starčević et al. confirmed that the regulation of NK cell activation in tumor invasion may be involved in the pathogenesis of breast cancer. The expression of peripheral NK cell-activated receptors (CD94/NKG2C, NKG2D, and CD16) and inhibitory receptors (NKG2A) from breast cancer patients was reduced, highlighting the importance of NK cells as suitable targets for effective antitumor responses in the breast cancer immunosuppressive tumor microenvironment⁹. Kruijf et al. confirmed that NKG2D ligand (NKG2DL) is often highly expressed in breast cancer, and NKG2DL can bind to the activating NKG2D (KLRK1) receptor present on NK cells and subsets of T cells, thus initiating immune response and playing a crucial role in tumor immune editing in breast cancer¹⁰. Other studies have linked NKG2D to the development of breast cancer¹¹⁻¹³. We have explained the reasons and references for adding these two markers in the revised manuscript (Line 194~201).

“Next, we applied a series of screening principles to identify the most important and specific DNA methylation markers of BC. By using LASSO analysis and $|\Delta\beta| \geq 0.08$, $p\text{-value} \leq 0.0001$ as the screening criteria, we obtained 6 methylation sites (Top 6 methylation sites in Table 1). It can be seen that most methylation sites are located in the Body region, and only cg11754974 is located in TSS1500. Given the notion that methylation in the TSS region has been shown to be more important than other CpG sites in regulating gene expression (31), we included 2 more methylation sites (the bottom 2 methylation sites in Table 1) in the TSS region that might be associated with the development of BC (32-37).” (Line 194~201)

31. Hong J & Rhee JK (2022) Genomic Effect of DNA Methylation on Gene Expression in Colorectal Cancer. *Biology (Basel)* 11(10).
32. Starcevic A, et al. (2022) Differences in tolerogenic status of NK cells between luminal A type, luminal B type, and triple-negative breast cancer. *Neoplasma* 69(6):1289-1302.
33. de Kruijf EM, et al. (2012) NKG2D ligand tumor expression and association with clinical outcome in early breast cancer patients: an observational study. *BMC Cancer* 12:24.
34. Varker KA, et al. (2007) Impaired natural killer cell lysis in breast cancer patients with high levels of psychological stress is associated with altered expression of killer immunoglobulin-like receptors. *J Surg Res* 139(1):36-44.
35. Mamessier E, et al. (2011) Human breast tumor cells induce self-tolerance mechanisms to avoid NKG2D-mediated and DNAM-mediated NK cell recognition. *Cancer Res* 71(21):6621-6632.
36. Raab S, et al. (2014) Fc-optimized NKG2D-Fc constructs induce NK cell antibody-dependent cellular cytotoxicity against breast cancer cells

- independently of HER2/neu expression status. *J Immunol* 193(8):4261-4272.
37. Verma C, *et al.* (2015) Natural killer (NK) cell profiles in blood and tumour in women with large and locally advanced breast cancer (LLABC) and their contribution to a pathological complete response (PCR) in the tumour following neoadjuvant chemotherapy (NAC): differential restoration of blood profiles by NAC and surgery. *J Transl Med* 13:180.

4. PBMCs cell composition: When working with PBMC, an essential parameter to take into consideration is the cell composition of the sample. Are the observed differences in DNA methylation due to a change affecting all cell populations or are they specific to a subpopulation? This question should be investigated using tools that allow for the evaluation of cell populations (e.g. MethyCIBERSORT).

Response: We agree and thank the reviewers for their constructive comments. According to the reviewer's suggestion, we analyzed each cellular component in PBMCs using EpiDISH method²⁰ published in *Nature methods*. we found significantly higher monocytes proportion in BC patients than in controls, and NK cells proportion in BC were significantly lower than those in normal controls, but not in CD8 + T cells, CD4 + T cells, B cells and granulocytes (**Figure 2 below and Revised Table S3 below**). Nevertheless, adjusted for the cell proportions, the eight CG sites we screened still showed significant difference between BC cases and controls (**Revised Table S4 below**). We have added the following statement as well as Revised Tables S3 and S4 to the results in the revised manuscript (Line 203~209).

“To rule out the influence of cell mixture distribution in PBMCs on DNA methylation, we calculated the cell proportion according to EpiDISH method (38), and found significantly higher monocytes proportion in BC patients than in controls, and NK cells proportion in BC were significantly lower than those in normal controls, but not in CD8 + T cells, CD4 + T cells, B cells and granulocytes (**Revised Table S3**). Nevertheless, adjusted for the cell proportions, the 8 CG sites we screened still showed significant difference between BC cases and controls (**Revised Table S4**).” (Line 203~209)

38. Zheng SC, Breeze CE, Beck S, & Teschendorff AE (2018) Identification of differentially methylated cell types in epigenome-wide association studies. *Nat Methods* 15(12):1059-1066.

Figure.2 The box plots present the distribution of cellular components in BC patients and normal controls. The X-axis represents each cell fraction. BC: breast cancer

Revised Table S3 Summary table of results of two-way ANOVA based on cell components

ID	Pr(>F) Group	Fvalues_Group	levene.Fvalue	levene.pvalue	shapino.Wvalue	shapino.pvalue	mean_BC	mean_Normal	sd_BC	sd_Normal
NK cells	0.000685938	12.50715707	1.532939929	0.219385399	0.979227545	0.218495904	0.233244137	0.307490116	0.082656365	0.103360388
Monocytes	0.005408004	8.188995686	3.315194167	0.072477333	0.94984011	0.003375059	0.185993509	0.150786642	0.056042147	0.048236214
Granulocytes	0.205399564	1.630664497	5.365943508	0.023160218	0.955310587	0.007026589	0.104854039	0.083931471	0.078846761	0.055081803
CD4+T cells	0.330760925	0.957826294	0.004804066	0.944919159	0.979370786	0.222993736	0.239582505	0.221635337	0.080228883	0.077995816
B cells	0.477784831	0.508806313	0.192740084	0.661859558	0.974276856	0.106444798	0.088582415	0.083257673	0.033174081	0.030833999
CD8+T cells	0.750400075	0.101912367	0.172116621	0.679376396	0.981844676	0.314824374	0.147743395	0.152898762	0.071118069	0.067867854

Revised Table S4 Table of EWAS results based on Logistic regression models.

probeID	Estimate	StdError	Zvalue	Pvalue
cg11754974	0.176546375	0.07317412	2.412688722	0.015835337
cg16652347	0.225803649	0.067394859	3.350458092	0.00080678
cg13828440	0.261032927	0.114833316	2.273146299	0.023017365
cg18637238	0.238724796	0.10023098	2.381746598	0.01723075
cg14507403	-0.060169757	0.018088697	-3.32637322	0.00087984
cg09821790	-0.061352485	0.030573082	-2.006748455	0.044776454
cg15694422	-0.257659442	0.070673533	-3.645769942	0.000266592
cg27527887	-0.141380207	0.045976006	-3.075086728	0.002104413

5. Line 234: “indicating that multiplex analysis was better than any single analysis (Figure 4A).” -> The authors should combine the results of the single analyses into a bioinformatic score to test whether that would improve the performance to the level of the multiplex analysis. When measured in individual assays, the markers could be weighted (weighted score using LASSO parameters) with respect to how much they contribute to the overall diagnostic performance. Indeed, by combining the 4 markers in one PCR assay instead of combining them bioinformatically by generating a score from 4 individual PCR assays greatly limits the possibilities to improve this assay or optimize it when it will be adapted to different clinical scenarios.

Response: Thanks for the reviewer's suggestion. Using LASSO analysis to weight individual markers can indeed improve performance to the level of multiple analyses (AUC=0.94). However, this study focuses on finding a more convenient, fast, and economical detection method. Multiple analysis can not only meet the clinical needs of fast and convenient detection, but also have high diagnostic performance

6. Figure 6B: In a cohort of healthy samples, using the BC-mqmsPCR assay, the authors predicted 17 positive BC cases among which two were later confirmed as BC. How about the other 15 positive BC cases? Although the short follow-up may partly explain why not more of these cases have been confirmed, it seems unlikely that all of them will develop BC. The authors should better discuss these results and particularly the potential high level of false positive cases of their assay and how this could be handled in the clinic.

Response: Thanks for the reviewer's suggestion. We believe that for the remaining 15 positive cases of breast cancer, with the extension of follow-up time, some patients may also be diagnosed with breast cancer, but some patients will not develop breast cancer, for the following reasons.

First, some studies have confirmed that DNA methylation has changed 4 years before cancer can be detected clinically²², so patients with positive BC-mqmsPCR test may have cancerous changes in their body, but the current clinical detection methods can't detect it, until the tumor develops to a certain size, it will be confirmed by clinical conventional detection methods; Secondly, the current clinical diagnosis of breast cancer mainly relies on imaging examinations, while the existing imaging examinations have their limitations. For example, mammography has a high false negative in patients

with dense breast, and the sensitivity of mammography decreases from 87% of fatty breast to 63% of dense breast^{23, 24}.

Breast ultrasound is highly dependent on the level of clinicians, and its value is limited for fat surrounded isoechoic lesions, heterogeneous echoic lesions surrounded by a heterogeneous background, deep lesions in huge breasts and subareolar lesions. Some breast tumors such as ductal carcinoma in situ and invasive lobular carcinoma are easily missed due to the nature of the lesions²⁵. Therefore, the patients who were wrongly grouped due to false negative of imaging examination were not excluded. Thirdly, BC-mqmsPCR detection method also has certain false positives, and this part of patients will not develop breast cancer with the extension of follow-up time.

Therefore, we recommend that patients with no risk factors for BC who test positive for BC-mqmsPCR should be reviewed with breast ultrasound or mammography every 6 months, and then 12 months and 24 months later. If the lesion remains stable, it can be rechecked every 1-2 years. Patients with high risk factors for BC who test positive for BC-mqmsPCR should be reviewed every 3 months. They were then reexamined at 6, 12 and 24 months. If the lesion remains stable, it can be rechecked every 1 year thereafter.

We have added cause discussion and clinical countermeasures in line 340~355 of the revised manuscript.

“We speculate that the above results may be caused by the following reasons. First, some patients with positive BC-mqmsPCR test may have cancerous changes in their body, but the current clinical detection methods can't detect it, until the tumor develops to a certain size, it will be confirmed by clinical conventional detection methods; Secondly, the current clinical diagnosis of BC mainly relies on imaging examinations, while the existing imaging examinations have their limitations (high false negative in patients with dense breast, depending on the level of clinicians, etc.), so patients wrongly grouped due to false negative imaging examination cannot be excluded. Thirdly, the detection method of BC-mqmsPCR also has certain false positives. With the extension of follow-up time, these patients will not develop BC.

Therefore, we recommend that patients with no risk factors for BC who test positive for BC-mqmsPCR should be reviewed with breast ultrasound or mammography every 6 months, and then 12 months and 24 months later. If the lesion remains stable, it can be rechecked every 1-2 years. Patients with high risk factors for BC who test positive for BC-mqmsPCR should be reviewed every 3 months. They were then reexamined at 6, 12 and 24 months. If the lesion remains stable, it can be rechecked every 1 year thereafter.” (Line 340~355)

Minor comments:

1. Include source that was used for SNP and CR filtering. Minor 1: SNP filtering source is not mentioned. Is it from the ChAMP package or another publication? Also, it is essential to check that the 4 markers do not overlap with cross-reactive probes (see e.g. McCartney et al).

Response: Thanks for the reviewer's suggestion. SNP filtering source are derived from the following studies^{26, 27}, which we have added to the references in the revised manuscript (Line 497). Yes, ChAMP packets are used to filter SNP. To check whether the 4 markers overlapped with cross-reaction probes, we performed multiple PCR using unmethylated human control DNA bisulfite converted (EpiTect® PCR Control DNA, QIAGEN, Cat. no.59695), which resulted in no amplification, demonstrating no cross-reaction (Table 1 below). In addition, the primers and probes of our 4 markers were specific and verified by TA cloning.

Table 1 Amplification results of multiple methylation specific PCR in unmethylated human control DNA (bisulfite converted) samples

Sample Name	Target Name	Reporter	CT
unmethylated human control DNA bisulfite converted	ACTB	VIC	30.549
unmethylated human control DNA bisulfite converted	BC-mqmsPCR	FAM	Undetermined
unmethylated human control DNA bisulfite converted	ACTB	VIC	30.357
unmethylated human control DNA bisulfite converted	BC-mqmsPCR	FAM	Undetermined
unmethylated human control DNA bisulfite converted	ACTB	VIC	30.431
unmethylated human control DNA bisulfite converted	BC-mqmsPCR	FAM	Undetermined

26. Zhou W, Laird PW, & Shen H (2017) Comprehensive characterization, annotation and innovative use of Infinium DNA methylation BeadChip probes. *Nucleic Acids Res* 45(4): e22.

27. Nordlund J, et al. (2013) Genome-wide signatures of differential DNA methylation in pediatric acute lymphoblastic leukemia. *Genome Biol* 14(9): r105.

2. Supp. Fig. 2A: PCA plot shows no difference between BC and normal controls. It is therefore misleading to state "to determine whether the methylation profile of BC patients differed from that of normal controls". It is better to describe the PCA as a simple control plot and to include a color code for the different hospitals to assess batch effect.

Response: We agree with the reviewer that this statement is inaccurate. We have revised the statement in line 173 of the revised manuscript. Because in the marker discovery stage, the 850k chip sequencing patients in cohort I come from a single center, so there is no batch effect.

“PCA plot shows little difference between BC and normal control.” (Line 173)

3. Supp. Fig. 2C: the enrichment mentioned by the author is far from obvious. Is the enrichment significant? And if it is, how can this be interpreted with respect to the overall message of this paper? Could it be due to CNV bias?

Response: Sorry that our description of this result is not accurate enough. We have modified the statement in line 183~184 of the revised manuscript.

“The distribution of DMPs varied among chromosomes.” (Line 183~184)

4. Rephrase “...the transcription start sites (TSS) of BC patients were more hypomethylated compared with normal controls” to “...the transcription start sites (TSS) of BC patients were more hypomethylated compared with normal controls”.

Response: Thank you! We have amended it in line 184~185 of the revised manuscript.

“Furthermore, the transcription start sites (TSS) of BC patients were more hypomethylated compared with normal controls (Figure. 2C).” (Line 184~185)

5. Visually, the delta-beta shown in Fig. 3A seem not to match the ones reported in Table 1. Please clarify and indicate the median in Fig. 3A.

Response: We thank the reviewer for this suggestion. The $\Delta\beta$ in Table 1 are the average β values of the BC group minus the average β values of the normal controls group. We have clarified and indicated the median in the revised manuscript of revised Figure 3A.

Revised Figures 3A. The violin plot shows the β -value distribution of 8 candidate methylation markers in PBMCs from BC patients (n = 50) and normal controls (n = 30). A β -value of zero represents no methylation, and 1 represents full methylation. The data are displayed as median with the interquartile range. *** $p \leq 0.001$; **** $p \leq 0.0001$.

6. The authors should not write “validate the 8 markers” while only 4 markers are actually validating. "Evaluate the reproducibility" would be a better fitting description.

Response: We thank the reviewer for this suggestion. We have revised the description in line 210~211 of the revised manuscript

“The reproducibility of these 8 markers was evaluated using pyrosequencing and TBS in a cohort of 200 patients.” (Line 210~211)

7. Supp. Fig. 3B: two of the final markers did not validate in TCGA. That is not surprising as TCGA is tumor tissue. This should be better highlighted/explained. In its current form, the TCGA analysis is rather confusing. Also, TCGA analysis needs to be described in Methods section.

Response: We thank the reviewer for this suggestion. Since all the data in TCGA were sequenced by 450K chip, excluding the two sites (cg11754974 and cg18637238) that we screened in 850K chip, it is impossible to evaluate the methylation differences between these two sites in breast cancer and adjacent normal tissues. The trend of methylation difference at site cg16652347 and cg13828440 in PBMCs is diametrically opposite to that in TCGA, which proves that the methylation difference we detected comes from PBMCs, not breast cancer tissue. We have added explanation in results section (line 215~218) and TCGA analysis in Methods section (line 542~550) of the revised manuscript.

“Notably, the methylation levels of cg16652347 and cg13828440 in the TCGA database were lower in BC tissues than in adjacent normal tissues, contrary to the trend of PBMCs, indicating differences in DNA methylation expression among different sample sources (Supplemental Figure. 3B).” (Line 215~218)

“Expression of candidate methylation markers in BC and adjacent normal tissues in TCGA database

In order to confirm the expression of four methylation markers screened from PBMCs in BC and normal tissues in TCGA database, we downloaded 450k Illumina Infinium methylation arrays sequencing data of 794 BC tissues and 96 adjacent normal tissues from TCGA database, and found that only two CpG sites cg16652347 and cg13828440 existed in the 450k methylation arrays sequencing data. Then, we compared the differences of DNA methylation expression of cg16652347 and cg13828440 in BC tissues and adjacent normal tissues by using rank sum test.” (Line 542~550)

8. Fig. 5J: Why is the detection rate of classical methods so low?

Response: Because the current routine use of serum tumor biomarkers CEA, CA153, and CA125 of breast cancer has low sensitivity, they may only be used for patients with advanced BC, which is not helpful for the screening or diagnosis of early BC^{28, 29}. For example, CA 153 concentrations are increased in 10% of patients with stage I disease, 20% with stage II disease, 40% with stage III disease, and 75% with stage IV disease^{30, 31}. Most of the cases we included were early breast cancer patients, so the detection rate of traditional tumor markers was very low.

9. Fig. 4I&J: Typo in “predicted”

Response: We apologize for this typo and have corrected it in the revised manuscript (Revised Figure 4I&J).

10. Fig. 5F: How was the cutoff defined? Is it always the same cutoff? Describe in main text, not just in Methods section.

Response: We apologize for the confusion. In the training set, we use the ROC curve to pass the corresponding value at the maximum value of the Youden’s index as the cut-off value. The same cut-off value is used in all subsequent analyses. We have added descriptions of cut-off value in main text (line 263~268) of the revised manuscript.

“To further assess the performance of BC-mqmsPCR assay for clinical application, we recruited a multicenter cohort of 206 BC patients and 170 normal controls from 10 hospitals in China and divided into training and validation sets. Training set (130 BC + 115 normal controls) were recruited from 6 centers for methodology development and search for the optimal cut-off value, and validation set (76 BC + 55 normal controls) were recruited from another 4 centers for external validation.” (263~268)

11. Fig. 6B: Not clear whether the assessment was made at t0 and/or at follow-up time?

Response: We apologize for the confusion. We defined the time point of BC-mqmsPCR detection after collecting samples from healthy controls as t0. Among 170 healthy controls patients, 17 patients were predicted to be positive and 153 patients were predicted to be negative. After 1-2 years of follow-up, 2 of the 17 patients were diagnosed with BC.

12. Fig. 3: Number of samples should be indicated in box plots.

Response: Thank you, we have replaced the box plots representation in the revised manuscript of Revised Figure 3.

Revised Figures 3A. The violin plot shows the β -value distribution of 8 candidate methylation markers in PBMCs from BC patients ($n = 50$) and normal controls ($n = 30$). A β -value of zero represents no methylation, and 1 represents full methylation. The data are displayed as median with the interquartile range. *** $p \leq 0.001$; **** $p \leq 0.0001$

13. Line 80: “This assay exhibited excellent performance in distinguishing early-stage BC patients, with an AUC of 0.940...” Distinguishing early-stage BC from what?

Response: We apologize for the confusion and have corrected it in line 79~80 of the revised manuscript.

“This assay exhibited excellent performance in distinguishing early-stage BC patients from normal controls, with an AUC of 0.940.” (Line 79~80)

14. Line 110/111: Change “DNA methylation exists in...” to “DNA methylation changes exist in...”

Response: Thank you. We have corrected it in line 110~111 of the revised manuscript.

15. Line 131: Change “...to identify BC-specific methylation landscape” to “...to identify BC-specific methylation landscape in PBMCs”

Response: Thank you. We have corrected it in line 130~131 of the revised manuscript.

16. Line 146: “low quality tests” -> Mention/descriptions of these tests is missing in Method section

Response: We thank the reviewer for this suggestion. We have added the description of “low quality tests” to the Method section (line 479~481) of the revised manuscript.

“Low quality tests (insufficient purity of PBMCs DNA after extraction, such as A260/280 \leq 1.6 or \geq 2.1, protein or RNA contamination, and cannot be accurately quantified) were removed.” (Line 479~481)

17. Line 154/155: Change “...identify BC-specific methylation markers” to “...identify BC-specific methylation markers in PBMCs”

Response: Thank you. We have corrected it in line 159~160 of the revised manuscript

18. Line 171: “p-value \leq 0.01” is different from the p-value mentioned in methods section for this analysis

Response: We apologize for the inconsistency and have replaced the method section in the revised manuscript with p-value \leq 0.01 (Line 504~505).

19. Line 203: Change “pyrophosphate” to “pyrosequencing”

Response: Thank you. We have revised it in the revised manuscript.

20. Line 204/205: Change “TCGA database showed hypomethylation levels in BC tissues” to “TCGA database showed hypomethylation in BC tissues”

Response: Thank you. We have revised it in the revised manuscript.

21. Line 207/208: “...the methylation level of cg18637238 was negatively correlated with tumor size...” -> What about cg16652347?

Response: Thank you. According to the pyrosequencing data in **Figure 3C**, the methylation level of cg16652347 in small tumor (\leq 1.5cm) is slightly higher than that in tumors $>$ 1.5cm, but there is no significant difference, while the methylation level of cg16652347 in early breast cancer is significantly higher than that in advanced breast cancer.

Figure 3C. The methylation levels of cg16652347 in normal controls and BC patients of the pyrosequencing set with different age, stage, tumor size, lymph node metastasis status, ER status, PR status, HER2 status and Ki-67 levels.

22. Suppl. Fig. 3C-F: TBS data look better than pyroseq data and could switched (TBS in main and pyroseq in suppl.)

Response: We thank the reviewer for this suggestion. We have adjusted the TBS data in main (**Revised Figure 3C-F**) and pyrosequencing in Supplementary (**Revised Supplementary Figure 3C-F**) in the revised manuscript.

23. Line 231/232: “ Δ CT values obtained from BC-mqmsPCR assay were significantly higher than...” -> Provide statistical test

Response: Sorry that our description is not accurate enough. We have revised the description in line 242~245 of the revised manuscript that "The BC-mqmsPCR assay produced a Δ CT value which was 2.22, 5.95, 0.71, and 1.76 higher than the individual cg11754974, cg16652347, cg13828440 and cg18637238 assays, respectively, indicating that the multiplex assay is analytically more sensitive than the individual assays".

24. Line 363 to 365; “These results suggest that the methylation changes in PBMCs may affect anti-tumor immune surveillance by regulating the expression of immune-related genes, which then impact the occurrence and progression of tumors.” -> Statement is too strong and speculative with respect to presented data

Response: We thank the reviewer for the helpful suggestions. We have revised the description in line 397 to 400 of the revised manuscript that "These results support the hypothesis that changes in DNA methylation in PBMC are associated with the immune system of the host organism and reflect the epigenetic reprogramming process of the immune system in BC".

25. Line 367 to 369: “our study found that the diagnostic efficacy of DNA methylation markers in PBMCs was higher for early-stage BC than for late-stage BC, and its performance was better for minimal tumors.” Incorrect statement as efficacy was compared between early-stage and late-stage BC but instead between early-stage and all-stage BC. Better to say “the diagnostic efficacy of DNA methylation markers in PBMCs was higher in a cohort of early-stage and minimal breast tumors than in a BC cohort of all stages and sizes”

Response: We thank the reviewer for the helpful suggestions. We have revised the description in line 401 to 403 of the revised manuscript that "our study found that the diagnostic efficacy of DNA methylation markers in PBMCs was higher in a cohort of early-stage and minimal breast tumors than in a BC cohort of all stages and sizes".

26. Figure 3/4: It would be interesting to see if the 4 final markers (Fig. 3) or the BC-mqmsPCR assay (Fig. 4) behave/perform different when BC cases are subgrouped into Lum A, Lum B, HER2 and TN subtypes. In other words, do the markers or the assay perform better in one BC subtype over others?

Response: We thank the reviewer for the helpful suggestions. Unfortunately, we did not find the 4 final markers (**Figure 3 below**) or the BC-mqmsPCR assay (**Figure 4 below**) perform different when BC cases are subgrouped into Lum A, Lum B, HER2

and TN subtypes.

Figure.3 The box plots present the distribution of 4 methylation markers in different breast cancer subtypes of pyrosequencing and TBS set.

Figure.4 The box plots present the distribution of BC-mqmsPCR assay in different breast cancer subtypes.

27. Figure 4I and 4J: What is the cut-off for positive/negative cases? How was the Cut-off defined? This assay is a key part of the study and should be described in more detail in the main text, not only in the Method section.

Response: We apologize for the confusion. The cut-off value of positive/negative cases in Figure 4I and 4J is 3.223. The cut-off value is defined as the critical value at the maximum value of the Youden's index by the ROC curve in the training set. We have added the description of cut-off values in lines 273~276 of the main text of the revised manuscript.

“More importantly, after determining the positive and negative classification of BC-mqmsPCR by the cutoff value of the Youden’s index (3.223) in the training set, we found that the predicted results of BC-mqmsPCR had a high consistency with the pathological diagnosis results (Figure 4I and J).” (Line 273~276)

Reviewer #3

(Remarks to the Author): expertise in breast cancer detection methods

This is an interesting manuscript. If further confirmatory studies are conducted, it may lead an impact. However, there may be some concerns.

Response: We appreciate your positive review and have added some further confirmatory studies based on your comments. We hope these experiments will address your concerns, and our responses to your questions are as follows.

High diagnostic performance seems to be indicated for Stage 0/DCIS. What is the detectability of premalignant lesions, benign breast diseases such as fibroadenoma and fibrocystic disease? Also, as briefly mentioned in the text, is there a difference between low-grade DCIS and high-grade DCIS?

Response: We appreciate the reviewer's suggestions. According to suggestions of reviewers, we collected 10 normal control cases, 12 benign breast lesions, 10 precancerous lesions, and 14 breast cancer cases. The DNA methylation level of these samples was simultaneously detected by BC-mqmsPCR method. It was found that the methylation degree of precancerous lesions and breast cancer patients was significantly higher than that of the normal control group, and there was no statistical significance between benign lesions and normal lesions (**Figure 5 below**). In addition, there was no significant difference in the degree of methylation between low-grade DCIS and high-grade DCIS (**Figure 6 below**).

Figure.5 BC-mqmsPCR analysis was performed on 10 normal controls, 12 benign breast diseases, 10 premalignant lesions and 14 breast cancer. ****P < 0.0001; ns, not significantly different.

Figure.6 The box plots present the degree of methylation of 4 methylation markers and BC-mqmsPCR assay in BC low-grade DCIS and high-grade DCIS.

What is the relationship between tumor subtype, grade, and tumor genomic instability/abnormalities? The relationship with pathological biomarkers needs to be summarized more clearly.

Response: We thank the reviewer for the helpful suggestions. Unfortunately, we did not find differences among the 4 final markers or BC-mqmsPCR assay across breast cancer subtypes (Figure 7 below) and grades (Figure 8 below).

Pyrosequencing

TBS

Figure.7 The box plots present the degree of methylation of 4 methylation markers and BC-mqmsPCR assay in different breast cancer subtypes.

Pyrosequencing

TBS

BC-mqmsPCR

Figure.8 The box plots present the degree of methylation of 4 methylation markers and BC-mqmsPCR assay in different breast cancer grade.

The association with tumour stage is indicated. What about that with nodal status, number of nodes involved? If there is a discrepancy in the relationship between tumor size and nodal status, it should be explained.

Response: We thank the reviewer for this suggestion. We analyzed the status of lymph node metastasis, and the results showed that the degree of differential methylation decreased with the increase of the number of lymph node metastases, which was consistent with the trend of tumor size and stage (**Figure 9 below**).

Figure.9 The box plot shows the degree of methylation at different N stages of breast cancer. N1: 1-3 lymph node metastases. N2: 4-9 lymph node metastases. N3: >9 lymph node metastases.

Is there an association between tumor infiltrating lymphocytes (TILs) and TIL features?

Response: We appreciate the reviewer's comments. Unfortunately, our study did not involve tumor infiltrating lymphocytes (TILs), so we were unable to determine a relationship with the TILs and TIL features. We will continue to explore the correlation in future studies.

Is there a relationship with profiles, such as the absolute number of lymphocytes in peripheral blood?

Response: We thank the reviewers for their constructive comments. We analyzed each cellular component in PBMCs using EpiDISH method²⁰ published in Nature methods. we found significantly higher monocytes proportion in BC patients than in controls, and NK cells proportion in BC were significantly lower than those in normal controls, but not in CD8 + T cells, CD4 + T cells, B cells and granulocytes (**Figure 10 below and Revised Table S3 below**). Nevertheless, adjusted for the cell proportions, the eight CG sites we screened still showed significant difference between BC cases and controls (**Revised Table S4 below**).

Figure.10 The box plots present the distribution of cellular components in BC patients and normal controls. The X-axis represents each cell fraction. BC: breast cancer

Revised Table S3 Summary table of results of two-way ANOVA based on cell components

ID	Pr(>F) Group	Fvalues_Group	levene.Fvalue	levene.pvalue	shapino.Wvalue	shapino.pvalue	mean_BC	mean_Normal	sd_BC	sd_Normal
NK cells	0.000685938	12.50715707	1.532939929	0.219385399	0.979227545	0.218495904	0.233244137	0.307490116	0.082656365	0.103360388
Monocytes	0.005408004	8.188995686	3.315194167	0.072477333	0.94984011	0.003375059	0.185993509	0.150786642	0.056042147	0.048236214
Granulocytes	0.205399564	1.630664497	5.365943508	0.023160218	0.955310587	0.007026589	0.104854039	0.083931471	0.078846761	0.055081803
CD4+T cells	0.330760925	0.957826294	0.004804066	0.944919159	0.979370786	0.222993736	0.239582505	0.221635337	0.080228883	0.077995816
B cells	0.477784831	0.508806313	0.192740084	0.661859558	0.974276856	0.106444798	0.088582415	0.083257673	0.033174081	0.030833999
CD8+T cells	0.750400075	0.101912367	0.172116621	0.679376396	0.981844676	0.314824374	0.147743395	0.152898762	0.071118069	0.067867854

Revised Table S4 Table of EWAS results based on Logistic regression models.

probeID	Estimate	StdError	Zvalue	Pvalue
cg11754974	0.176546375	0.07317412	2.412688722	0.015835337
cg16652347	0.225803649	0.067394859	3.350458092	0.00080678
cg13828440	0.261032927	0.114833316	2.273146299	0.023017365
cg18637238	0.238724796	0.10023098	2.381746598	0.01723075
cg14507403	-0.060169757	0.018088697	-3.32637322	0.00087984
cg09821790	-0.061352485	0.030573082	-2.006748455	0.044776454
cg15694422	-0.257659442	0.070673533	-3.645769942	0.000266592
cg27527887	-0.141380207	0.045976006	-3.075086728	0.002104413

What about the performance for invasive lobular carcinoma?

Response: There was no significant difference in the degree of methylation between invasive lobular carcinoma and other types of breast cancer (**Figure 11 below**).

Figure.11 The box plot shows the degree of methylation in different pathological types of breast cancer. IDC: Invasive ductal carcinoma. ILC: Invasive lobular carcinoma. NIC: Non-invasive cancer. ISC: Invasive specific carcinoma.

What about time-course/ follow-up data after surgery and also after anti-tumor therapy? Is it possible to show changes of abnormalities that authors focused in this study with respect to the magnitude of abnormalities and also duration of the abnormalities? I would like to know the half-life of this event, if any.

Response: We appreciate the reviewer's very constructive suggestions. Unfortunately, we did not collect samples of patients after surgery or anti-tumor therapy, so the follow-up data of these patients is missing. In the future, we plan to collect samples of some patients at different time interval points after surgery and after anti-tumor therapy to study whether the methylation changes concerned in this study will change with patients after treatment and how long the interval between changes will be.

The difference between LC, GC, and CRC is clearly shown, but the mechanism of difference needs to be explained a little more.

Response: We thank the reviewer for this suggestion. Cancer has specific antigens, and some of these differences are recognized by the immune system³². Intrinsic characteristics of tumors have been shown to influence the immune response of different cancer types³³. Zhang et al. clarified in a study on HCC-related DNA methylation markers in PBMCs that the observed methylation changes were unique in patients with HCC. Because HCC is a disease often caused by chronic inflammation of the liver, however, compared with the effect of chronic hepatitis on DNA methylation, the change of DNA methylation in HCC is significantly enhanced⁴. Moreover, the verified methylation markers of PBMCs in HCC are different from those of breast cancer in our study, so we speculate that different tumors can trigger host-specific immune responses. The specific mechanisms involved will be discussed in subsequent studies.

Minor comments:

- Inter-institutional sample quality control/assurance should be addressed.

Response: We thank the reviewer for this suggestion. In order to ensure the quality of multi-center samples and reduce experimental errors, we developed unified sample collection standards in multi-center hospitals, and then all collected samples were uniformly mailed to the same laboratory for testing. After the quality inspection of specimens, the samples that meet the requirements are included in the study and then tested by a specially-assigned person.

- The data in Figure 6, especially the follow-up data, are very preliminary and maybe too early to present. It might be better to show it in suppl.

Response: We strongly agree with the reviewer's suggestion, and we have shown Figure 6 in the Supplementary Figure 6 in the revised manuscript.

1. Hiam-Galvez K. J., Allen B. M. & Spitzer M. H. Systemic immunity in cancer. *Nat Rev Cancer*. **21**, 345-359 (2021).
2. Huang W. Y., et al. Prospective study of genomic hypomethylation of leukocyte DNA and colorectal cancer risk. *Cancer Epidemiol Biomarkers Prev*. **21**, 2014-21 (2012).
3. Mehdi A., et al. DNA methylation signatures of Prostate Cancer in peripheral T-cells. *BMC Cancer*. **20**, 588 (2020).
4. Zhang Y., et al. The signature of liver cancer in immune cells DNA methylation. *Clin Epigenetics*. **10**, 8 (2018).
5. Wolf N. K., Kissiov D. U. & Raulet D. H. Roles of natural killer cells in immunity to cancer, and applications to immunotherapy. *Nat Rev Immunol*. **23**, 90-105 (2023).
6. Jin S., et al. Efficient detection and post-surgical monitoring of colon cancer with a multi-marker DNA methylation liquid biopsy. *Proc Natl Acad Sci U S A*. **118**, (2021).
7. Hong J. & Rhee J. K. Genomic Effect of DNA Methylation on Gene Expression in

- Colorectal Cancer. *Biology (Basel)*. **11**, (2022).
8. Varker K. A., et al. Impaired natural killer cell lysis in breast cancer patients with high levels of psychological stress is associated with altered expression of killer immunoglobulin-like receptors. *J Surg Res*. **139**, 36-44 (2007).
 9. Starcevic A., et al. Differences in tolerogenic status of NK cells between luminal A type, luminal B type, and triple-negative breast cancer. *Neoplasma*. **69**, 1289-1302 (2022).
 10. de Kruijf E. M., et al. NKG2D ligand tumor expression and association with clinical outcome in early breast cancer patients: an observational study. *BMC Cancer*. **12**, 24 (2012).
 11. Mamessier E., et al. Human breast tumor cells induce self-tolerance mechanisms to avoid NKG2D-mediated and DNAM-mediated NK cell recognition. *Cancer Res*. **71**, 6621-32 (2011).
 12. Raab S., et al. Fc-optimized NKG2D-Fc constructs induce NK cell antibody-dependent cellular cytotoxicity against breast cancer cells independently of HER2/neu expression status. *J Immunol*. **193**, 4261-72 (2014).
 13. Verma C., et al. Natural killer (NK) cell profiles in blood and tumour in women with large and locally advanced breast cancer (LLABC) and their contribution to a pathological complete response (PCR) in the tumour following neoadjuvant chemotherapy (NAC): differential restoration of blood profiles by NAC and surgery. *J Transl Med*. **13**, 180 (2015).
 14. Krammer P. H., Behrmann I., Daniel P., Dhein J. & Debatin K. M. Regulation of apoptosis in the immune system. *Curr Opin Immunol*. **6**, 279-89 (1994).
 15. Nagata S. & Tanaka M. Programmed cell death and the immune system. *Nat Rev Immunol*. **17**, 333-340 (2017).
 16. Jellusova J. Metabolic control of B cell immune responses. *Curr Opin Immunol*. **63**, 21-28 (2020).
 17. Tang Q., et al. DNA methylation array analysis identifies breast cancer associated RPTOR, MGRN1 and RAPSN hypomethylation in peripheral blood DNA. *Oncotarget*. **7**, 64191-64202 (2016).
 18. Zhu H., et al. Rheumatoid arthritis-associated DNA methylation sites in peripheral blood mononuclear cells. *Ann Rheum Dis*. **78**, 36-42 (2019).
 19. Chaix R., et al. Differential DNA methylation in experienced meditators after an intensive day of mindfulness-based practice: Implications for immune-related pathways. *Brain Behav Immun*. **84**, 36-44 (2020).
 20. Zheng S. C., Breeze C. E., Beck S. & Teschendorff A. E. Identification of differentially methylated cell types in epigenome-wide association studies. *Nat Methods*. **15**, 1059-1066 (2018).
 21. Wu X., et al. A novel cell-free DNA methylation-based model improves the early detection of colorectal cancer. *Mol Oncol*. **15**, 2702-2714 (2021).
 22. Chen X., et al. Non-invasive early detection of cancer four years before conventional diagnosis using a blood test. *Nat Commun*. **11**, 3475 (2020).
 23. Vourtsis A. & Berg W. A. Breast density implications and supplemental screening. *Eur Radiol*. **29**, 1762-1777 (2019).

24. Carney P. A., et al. Individual and combined effects of age, breast density, and hormone replacement therapy use on the accuracy of screening mammography. *Ann Intern Med.* **138**, 168-75 (2003).
25. Uematsu T. Ultrasonographic findings of missed breast cancer: pitfalls and pearls. *Breast Cancer.* **21**, 10-9 (2014).
26. Zhou W., Laird P. W. & Shen H. Comprehensive characterization, annotation and innovative use of Infinium DNA methylation BeadChip probes. *Nucleic Acids Res.* **45**, e22 (2017).
27. Nordlund J., et al. Genome-wide signatures of differential DNA methylation in pediatric acute lymphoblastic leukemia. *Genome Biol.* **14**, r105 (2013).
28. Mirabelli P. & Incoronato M. Usefulness of traditional serum biomarkers for management of breast cancer patients. *Biomed Res Int.* **2013**, 685641 (2013).
29. Lumachi F., et al. Long-term follow-up study in breast cancer patients using serum tumor markers CEA and CA 15-3. *Anticancer Res.* **19**, 4485-9 (1999).
30. Duffy M. J., Evoy D. & McDermott E. W. CA 15-3: uses and limitation as a biomarker for breast cancer. *Clin Chim Acta.* **411**, 1869-74 (2010).
31. Duffy M. J. Serum tumor markers in breast cancer: are they of clinical value? *Clin Chem.* **52**, 345-51 (2006).
32. Srivastava P. K. Neoepitopes of Cancers: Looking Back, Looking Ahead. *Cancer Immunol Res.* **3**, 969-77 (2015).
33. Spranger S. & Gajewski T. F. Impact of oncogenic pathways on evasion of antitumour immune responses. *Nat Rev Cancer.* **18**, 139-147 (2018).

REVIEWERS' COMMENTS

Reviewer #1 (Remarks to the Author):

I thank the authors for their response to review. While they have addressed the majority of my concerns, I still have several two minor review queries:

- Review point 3: R is not the package, it's the programming language, so the package used for gene set enrichment is still unknown meaning the results can't be reproduced. Furthermore, the details of this and the Pyro Q-CPG software need added to the methods section of the paper and not just the response to reviewer.
- C-mqmsPCR Probability score is not a probability (it is not even between 0 and 1), so why is it named as such? Can it be renamed to reflect what it actually is (i.e. a delta ct)?

Reviewer #2 (Remarks to the Author):

Overall, the revision of "A Multiplex Blood-Based Assay Targeting DNA methylation in PBMCs Enables Early Detection of Breast Cancer", by Wang et al., led to improvements of the manuscript. We appreciate the efforts made by Wang and colleagues to reply to all of our comments and to follow most of our suggestions. The revised manuscript appears more clear, but a few issues remain:

Major comment #3:

We thank the authors for the clear description of the selection strategy to us. The text they added to the revised manuscript (line 194-201), however, is not as clear as what they described in their reply to this comment. We thus suggest that they revise the text line 194-201 in a way that it is as clear as stated in the text added to line 508-521 that they applied three different selection principles. We further suggest that they clarify that the $|\Delta\beta| \geq 0.08$, $p\text{-value} \leq 0.0001$ screening principle is based on the Champ DMP function. Moreover, in the text from line 194-201, the authors should add some of the explanations they provided to us in their reply regarding the involvement in the development of breast cancer of the two methylation sites that were additionally added. We refer to this text "Varker et al. showed that cancer-related stress in patients with invasive breast cancer is related to immune impairment. In patients exhibiting high levels of cancer-related psychological stress, NK cytotoxicity is reduced and CD94 (KLRD1) levels are reduced⁸. Starčević et al. confirmed that the regulation of NK cell activation in tumor invasion may be involved in the pathogenesis of breast cancer. The expression of peripheral NK cell-activated receptors (CD94/NKG2C, NKG2D, and CD16) and inhibitory receptors (NKG2A) from breast cancer patients was reduced, highlighting the importance of NK cells as suitable targets for effective antitumor responses in the breast cancer immunosuppressive tumor microenvironment⁹. Kruijf et al. confirmed that NKG2D ligand (NKG2DL) is often highly expressed in breast cancer, and NKG2DL can bind to the activating NKG2D (KLRK1) receptor present on NK cells and subsets of T cells, thus initiating immune response and playing a crucial role in tumor immune editing in breast cancer¹⁰. Other studies have linked NKG2D to the development of breast cancer¹¹⁻¹³. We have explained the reasons and references for adding these two markers in the revised manuscript (Line 194~201)."

Major comment #4:

We thank the authors for addressing our comment regarding the PBMCs cell composition. While their analysis suggested that the cell composition alone is not sufficient to explain the differences in DNA methylation measured by the 8 CG sites, we wonder what does this signature reflect then? Could the authors speculate a bit about this in the discussion?

Major comment #5:

The reply the authors made to this comment should be included in the main manuscript. Add the result of combined Lasso (i.e. (AUC=0.94)) to line 252-253 as well as the explanation why you chose to continue with the mqmsPCR analysis.

Major comment #6:

The discussion added to the revised manuscript (line 340-355), is too short with respect to the possibility that the BC-mqmsPCR test has a high rate of false positives (Supplemental Fig. 6B indicates that this may be the case). The authors need to discuss follow-up studies, which need to be carried out to determine the exact false positive rate before they talk about screening recommendations based on this test.

Minor comment #4:

Rephrase "...the transcription start sites (TSS) of BC patients were more hypomethylated compared with normal controls" to "...the transcription start sites (TSS) of BC patients were hypomethylated compared with normal controls".

Minor comment #5:

Describe in the legend of Table 1 what you explained to us in your reply to this comment (i.e. it is the average, not the median).

Minor comment #7:

The authors clarified well to us in their reply what the aim of using TCGA data was, but they do not explain it in the manuscript. Thus, please add this explanation to the results section to give the reader the chance to understand.

Minor comment #8:

The explanation given to us in their reply to this comment should be added to the main manuscript.

Minor comment #11:

The explanation given to us in their reply to this comment is still not clear. Please clarify better and add the clarification to the Methods section and legend of the figure.

Minor comment #23:

We asked the authors to provide a statistical test for the results described in line 242~245 of the revised manuscript. The results of the tests are still missing.

Figure 3E-F and Suppl. Fig. 3E-F:

Where is the unsupervised hierarchical clustering mentioned in the text (line 230-232)? The displayed heatmap is supervised.

Figure 4A:

deltaRN in figure but deltaCt in text.

Reviewer #3 (Remarks to the Author):

The authors have accurately responded point by point and are generally well addressed. Some items were impossible to analyse, for instance because of lack of samples or of technology, or commented on as negative outcomes. These issues are described mostly in the text, but since it is also limitation in this study, it may be better to describe it a little bit more clearly.

Response to reviewer's comments:

We are pleased to resubmit for publication a revised version of this manuscript. We thank the reviewers for their positive comments and suggestions. Following the reviewers' advice, we have corrected the errors in the manuscript, added comments and clarifications to the manuscript text as suggested by the reviewers (see below for details), and highlighted all the changes in the text in red, hoping that these will greatly enhance the submission. We have also responded below:

Reviewer #1

I thank the authors for their response to review. While they have addressed the majority of my concerns, I still have several two minor review queries:

- Review point 3: R is not the package, it's the programming language, so the package used for gene set enrichment is still unknown meaning the results can't be reproduced. Furthermore, the details of this and the Pyro Q-CPG software need added to the methods section of the paper and not just the response to reviewer.

Response: Thank the reviewer for this suggestion. We have added the details of the package for gene enrichment (line 555-571) and Pyro Q-CPG software (line 591~593) to the Methods section of the revised manuscript.

“Gene ontology (GO) analysis and kyoto encyclopedia of genes and genomes (KEGG) analyses

To clarify the biological functions of the genes and the involved signaling pathways, we annotated each Gene based on the GO and KEGG database. Enrichment calculations were performed using Fisher's exact test. Further, we also need to conduct GO and pathway enrichment analysis of the genes. The specific principle is to carry out annotation mapping of differentially expressed genes in GO and KEGG database entries, calculate the number of the genes in each GO and pathway entry, and then use hypergeometric test for statistics. Select the GO and KEGG entries that are significantly enriched in the differentially expressed genes. After the calculated p-value was corrected by multiple hypothesis tests, the P-value 0.05 was taken as the threshold, and the GO and KEGG term meeting this condition was defined as the GO and KEGG term

significantly enriched in the target genes, Rich Factor, whose calculation formula is: $(\text{diff_gene_in_this_pathway}/\text{diff_gene_in_all_pathway})$ $(\text{all_gene_in_this_pathway}/\text{all_gene_in_all_pathway})$. The biological processes of GO and KEGG pathways enrichment analyses were carried out by clusterProfiler package of R. The figure was drawn by ggplot2". (Line 555-571)

"The absolute methylation levels of CpG sites were calculated by Pyro Q-CpG Software⁷⁰ that calculates the ratio of converted to unconverted cytosines at each CpG". (Line 591~593)

70. England R. & Pettersson M. Pyro Q-CpGTM: quantitative analysis of methylation in multiple CpG sites by Pyrosequencing®. *Nat Methods*. (2005).

- BC-mqmsPCR Probability score is not a probability (it is not even between 0 and 1), so why is it named as such? Can it be renamed to reflect what it actually is (i.e. a delta ct)?

Response: Thank you for this comment. We have renamed it to BC-mqmsPCR value (Δ CT) and modified it in the revised manuscript of Figure 4K.

Reviewer #2

Overall, the revision of "A Multiplex Blood-Based Assay Targeting DNA methylation in PBMCs Enables Early Detection of Breast Cancer", by Wang et al., led to improvements of the manuscript. We appreciate the efforts made by Wang and colleagues to reply to all of our comments and to follow most of our suggestions. The revised manuscript appears more clear, but a few issues remain:

Major comment #3:

We thank the authors for the clear description of the selection strategy to us. The text they added to the revised manuscript (line 194-201), however, is not as clear as what they described in their reply to this comment. We thus suggest that they revise the text line 194-201 in a way that it is as clear as stated in the text added to line 508-521 that they applied three different selection principles. We further suggest that they clarify that

the $|\Delta\beta| \geq 0.08$, $p\text{-value} \leq 0.0001$ screening principle is based on the Champ DMP function. Moreover, in the text from line 194-201, the authors should add some of the explanations they provided to us in their reply regarding the involvement in the development of breast cancer of the two methylation sites that were additionally added. We refer to this text “Varker et al. showed that cancer-related stress in patients with invasive breast cancer is related to immune impairment. In patients exhibiting high levels of cancer-related psychological stress, NK cytotoxicity is reduced and CD94 (KLRD1) levels are reduced⁸. Starčević et al. confirmed that the regulation of NK cell activation in tumor invasion may be involved in the pathogenesis of breast cancer. The expression of peripheral NK cell-activated receptors (CD94/NKG2C, NKG2D, and CD16) and inhibitory receptors (NKG2A) from breast cancer patients was reduced, highlighting the importance of NK cells as suitable targets for effective antitumor responses in the breast cancer immunosuppressive tumor microenvironment⁹. Kruijf et al. confirmed that NKG2D ligand (NKG2DL) is often highly expressed in breast cancer, and NKG2DL can bind to the activating NKG2D (KLRK1) receptor present on NK cells and subsets of T cells, thus initiating immune response and playing a crucial role in tumor immune editing in breast cancer¹⁰. Other studies have linked NKG2D to the development of breast cancer¹¹⁻¹³. We have explained the reasons and references for adding these two markers in the revised manuscript (Line 194~201).”

Response: Thank you for this comment. According to the reviewer's suggestion, we have revised line 194-201 and added explanations for the selection of two other methylation sites involved in the development of breast cancer in the revised manuscript (Line 193~219).

“Next, we applied a series of screening principles to identify the most important and specific DNA methylation markers of BC. By using LASSO analysis, 33 methylation markers were obtained (Table S3), and using $|\Delta\beta| \geq 0.08$, $p\text{-value} \leq 0.0001$ screening principle (based on the Champ DMP function), 8 methylation markers were obtained (Table S4), with 6 methylation markers overlapped in the two screening criteria. After removing 2 methylation markers (cg21723696 and cg14928964) that were difficult to design with primers, we ultimately selected 4 overlapping methylation markers

(cg14507403, cg09821790, cg15694422 and cg27527887) and 2 methylation markers (cg11754974 and cg16652347) that only met $|\Delta\beta| \geq 0.08$, $p\text{-value} \leq 0.0001$ screening principle for further validation. Given the notion that methylation in the TSS region has been shown to be more important than other CpG sites in regulating gene expression¹, we included 2 more methylation sites (cg13828440/KLRD1, cg18637238/KLRK1) in the TSS region that might be associated with the development of BC. Varker et al. showed that cancer-related stress in patients with invasive BC is related to immune impairment. In patients exhibiting high levels of cancer-related psychological stress, NK cytotoxicity is reduced and CD94 (KLRD1) levels are reduced². Starčević et al. confirmed that the regulation of NK cell activation in tumor invasion may be involved in the pathogenesis of BC. The expression of peripheral NK cell-activated receptors (CD94/NKG2C, NKG2D, and CD16) and inhibitory receptors (NKG2A) from BC patients was reduced, highlighting the importance of NK cells as suitable targets for effective antitumor responses in the BC immunosuppressive tumor microenvironment³. Kruijf et al. confirmed that NKG2D ligand (NKG2DL) is often highly expressed in BC, and NKG2DL can bind to the activating NKG2D (KLRK1) receptor present on NK cells and subsets of T cells, thus initiating immune response and playing a crucial role in tumor immune editing in BC⁴. Other studies have linked NKG2D to the development of BC⁵. Therefore, a total of 8 candidate markers were selected for the final validation, including 4 hyper-methylated markers and 4 hypo-methylated markers (Table 1, Fig. 2E, 3A).” (Line 193~219)

Major comment #4:

We thank the authors for addressing our comment regarding the PBMCs cell composition. While their analysis suggested that the cell composition alone is not sufficient to explain the differences in DNA methylation measured by the 8 CG sites, we wonder what does this signature reflect then? Could the authors speculate a bit about this in the discussion?

Response: Thank the reviewer for this suggestion. We have added some speculation about this possibility in the discussion section of the revised manuscript (Line 418~428).

“Differences in DNA methylation profiles might be influenced by the proportions of the PBMCs composition. According to our Illumina 850K data, the proportion of Monocyte in BC patients was higher than that in normal controls, and the proportion of NK cells was lower than that in normal controls, while the proportion of CD8+T cells, CD4+T cells, B cells and granulocytes was similar between BC patients and normal controls. After adjusting for cell proportion, the eight markers we screened still showed significant differences between BC and normal controls, suggesting that changes in the proportion of PBMCs subsets were not the reason for the observed difference in methylation in PBMCs, and that changes in methylation profile may be the activation of the host immune responses during BC progression. The differentially methylated genes were enriched in immune functions, such as regulation of B cell proliferation, regulation of myeloid leukocyte differentiation, and regulation of lymphocyte apoptotic process, supporting the hypothesis that DNA methylation changes in PBMCs of BC patients are associated with host immune responses”. (Line 418~428)

Major comment #5:

The reply the authors made to this comment should be included in the main manuscript. Add the result of combined Lasso (i.e. (AUC=0.94)) to line 252-253 as well as the explanation why you chose to continue with the mqmsPCR analysis.

Response: We appreciate the reviewer's suggestions and have added the results of combined Lasso to the revised manuscript and explained why we chose mqmsPCR analysis (Line 275-279).

“Although using LASSO analysis to weight individual markers can also improve performance to the level of multiple analyses (AUC=0.94). However, this study focuses on finding a more convenient, fast, and economical detection method. Multiple analysis can not only meet the clinical needs of rapid and convenient clinical detection, but also has high diagnostic efficiency”. (Line 275-279)

Major comment #6:

The discussion added to the revised manuscript (line 340-355), is too short with respect to the possibility that the BC-mqmsPCR test has a high rate of false positives (Supplemental Fig. 6B indicates that this may be the case). The authors need to discuss

follow-up studies, which need to be carried out to determine the exact false positive rate before they talk about screening recommendations based on this test.

Response: We thank the reviewers for their suggestions and have added a discussion on the possibility of false positive rates of BC-mqmsPCR testing to the revised manuscript and given corresponding screening recommendations (Line 378-384).

“Thirdly, the detection method of BC-mqmsPCR also has certain false positives. Cross-contamination of target sequences or amplification products is an important factor leading to false positives, among which aerosol contamination is the most common. Secondly, cross-contamination of PCR reagents and samples can also lead to false positives. In this regard, we should standardize the laboratory design and complete different experimental operations in continuous independent space. Regular ventilation and disinfection, and the use of aerosol remover. Use disposable utensils or autoclaved consumables to avoid cross-contamination. With the extension of follow-up time, these patients will not develop BC”. (Line 378-384))

Minor comment #4:

Rephrase “...the transcription start sites (TSS) of BC patients were more hypomethylated compared with normal controls” to “...the transcription start sites (TSS) of BC patients were hypomethylated compared with normal controls”.

Response: Thank you! We have amended it in line 183~184 of the revised manuscript. “Furthermore, the transcription start sites (TSS) of BC patients were hypomethylated compared with normal controls.” (Line 183~184)

Minor comment #5:

Describe in the legend of Table 1 what you explained to us in your reply to this comment (i.e. it is the average, not the median).

Response: We thank the reviewer for this suggestion. We have added a note to the legend in Table 1 of the revised manuscript.

“*The $\Delta\beta$ value (BC-NC) is the average β values of the BC group minus the average β values of the normal controls group”.

Minor comment #7:

The authors clarified well to us in their reply what the aim of using TCGA data was, but they do not explain it in the manuscript. Thus, please add this explanation to the results section to give the reader the chance to understand.

Response: We thank the reviewer for this suggestion. We have added explanations related to TCGA data analysis in the results section of the revised manuscript (Line 232-238).

“Notably, the trend of methylation difference at site cg16652347 and cg13828440 in TCGA is diametrically opposite to that in PBMCs, which proves that the methylation difference we detected comes from PBMCs, not BC tissue (Supplemental Fig. 3B). Since all the data in TCGA were sequenced by 450K chip, excluding the two sites (cg11754974 and cg18637238) that we screened in 850K chip, it is impossible to evaluate the methylation differences between these two sites in BC and adjacent normal tissues”. (Line 232-238)

Minor comment #8:

The explanation given to us in their reply to this comment should be added to the main manuscript.

Response: We thank the reviewer for this suggestion. We have added explanations of the low detection rate of traditional tumor markers in the results section of the revised manuscript (Line 344-349).

“Because the traditional tumor markers CEA, CA153, and CA125 may only be used in patients with advanced BC, they are not helpful for screening or diagnosis of early BC^{41, 42}. For example, CA153 concentrations are increased in 10% of patients with stage I disease, 20% with stage II disease, 40% with stage III disease, and 75% with stage IV disease^{43, 44}, so the detection rate of traditional tumor markers in early BC is very low.” (Line 344-349)

Minor comment #11:

The explanation given to us in their reply to this comment is still not clear. Please clarify better and add the clarification to the Methods section and legend of the figure.

Response: We apologize for the confusion. We defined the time point of BC-

mqmsPCR detection after collecting samples from normal controls as Test time 0. As can be seen from the figure, among 170 normal controls at Test time 0, 17 were predicted to be positive and 153 were predicted to be negative. Then we followed up the 170 normal controls by telephone, with an average follow-up every six months. The last follow-up time minus the collection time was defined as the follow-up time of each sample. As can be seen from the figure, after 1-2 years of follow-up, 2 of the 17 patients predicted to be positive were diagnosed with BC. However, among the 153 predicted negative patients, no BC was confirmed within 2 years of follow-up. We have added explanations in the Methods section (Line 632-642) and legend of the Supplementary figure 6 in the revised manuscript.

“Supplementary Fig. 6 Follow-up results of 170 normal controls. (a) The follow-up results of 170 normal controls. **(b)** Scatter plots of follow-up results of 170 normal controls predicted by BC-mqmsPCR. Test time 0 was defined as the BC-mqmsPCR detection time during specimen collection, and follow-up time was defined as the last follow-up time minus specimen collection time. Among the 15 cases with positive prediction, two were diagnosed as breast cancer within 1 year of follow-up; and among the 131 cases with negative prediction, no confirmed breast cancer was found within 2 years of follow-up. Red hollow circles represent patients with positive prediction who were not diagnosed with breast cancer; blue hollow circles represent patients with negative prediction who were not diagnosed with breast cancer; red solid star represent patients with positive prediction who were diagnosed with breast cancer”.

“Follow up process

We conducted telephone follow-up on 170 normal controls included in the training and validation sets. Firstly, we defined the time point of BC-mqmsPCR detection after collecting samples from normal controls as Test time 0. According to the cut-off value of 3.223, of 170 normal controls, 17 predicted positive and 153 predicted negative. Then we followed up every six months for 2 years on average. The last follow-up time minus the collection time was defined as the follow-up time of each sample. During the follow-up, pathological examination was carried out for those with abnormal imaging

examination. According to the pathological results, it was confirmed whether the patient had developed BC. No pathological examination was performed in patients with no abnormal imaging findings during follow-up". (Line 632-642)

Minor comment #23:

We asked the authors to provide a statistical test for the results described in line 242~245 of the revised manuscript. The results of the tests are still missing.

Response: I'm sorry our description wasn't accurate enough. What we want to express here is that multiplex assay is the accumulation of fluorescence signals at a single CpG site. It can be seen from the figure that the BC-mqmsPCR assay produced a Δ CT value which was 2.22, 5.95, 0.71, and 1.76 higher than the individual cg11754974, cg16652347, cg13828440 and cg18637238 assays, respectively. In order to describe the relevant results more accurately, we have revised the relevant description to "The BC-mqmsPCR assay produced a Δ CT value which was 2.22, 5.95, 0.71, and 1.76 higher than the individual cg11754974, cg16652347, cg13828440 and cg18637238 assays, respectively, confirming that multiplex assay could achieve fluorescence signal accumulation and is analytically more sensitive than the individual assays". (Line 262-266)

Figure 3E-F and Suppl. Fig. 3E-F:

Where is the unsupervised hierarchical clustering mentioned in the text (line 230-232)?
The displayed heatmap is supervised.

Response: We apologize for this typo and have corrected it in the revised manuscript (Revised Figure 3E-F and Suppl. Fig. 3E-F:)

Figure 4A:

deltaRN in figure but deltaCt in text.

Response: We apologize for the confusion. The value of Δ Rn represents the amount of degradation of the probe during the PCR process, which is the amount of PCR product. Δ Rn=R+n - R-n, where R+n represents the fluorescence intensity measured at each point, and R-n represents the fluorescence baseline intensity. The value of Δ Rn is used

to curve the number of cycles and select a fluorescence Threshold. The number of cycles in which fluorescence reaches the fluorescence threshold is called the Cycle Threshold (Ct), and the Ct value decreases linearly with the increase of the amount of these initial templates, so that the amount of the initial template can be calculated from the Ct value. The ΔCT value represents the methylation level of the marker ($\Delta CT = CT_{\text{reference}} - CT_{\text{biomarker}}$). We have added explanations in the legend of the figure (Line 923-926) in the revised manuscript.

“The value of ΔR_n represents the amount of degradation of the probe during the PCR process, which is the amount of PCR product. $\Delta R_n = R_{+n} - R_{-n}$, where R_{+n} represents the fluorescence intensity measured at each point, and R_{-n} represents the fluorescence baseline intensity”. (Line 923-926)

Reviewer #3

The authors have accurately responded point by point and are generally well addressed. Some items were impossible to analyse, for instance because of lack of samples or of technology, or commented on as negative outcomes.

These issues are described mostly in the text, but since it is also limitation in this study, it may be better to describe it a little bit more clearly.

Response: We thank the reviewer for this suggestion. We have added the shortcomings of the study to the limitations of the revised manuscript (Line 485-486).

“In the next study, we will further explore the relevant issues that cannot be analyzed in this study (Line 485-486)”.